# The full transcription map of cottontail rabbit papillomavirus in tumor tissues

**Pengfei Jiang**[1¤◉], **Vladimir Majerciak**[1◉], **Jiafen Hu**[2◉], **Karla Balogh**[2], **Thomas J. Meyer**[3], **Maggie Cam**[3], **Debra Shearer**[2], **Matthew Lanza**[4], **Neil D. Christensen**[2]*, **Zhi-Ming Zheng**[1]*

1 Tumor Virus RNA Biology Section, The HIV Dynamics and Replication Program, NCI, NIH, Frederick, Maryland, United States of America, 2 The Jake Gittlen Laboratories for Cancer Research, Department of Pathology and Laboratory Medicine, Pennsylvania State University College of Medicine, Hershey, Pennsylvania, United States of America, 3 CCR Collaborative Bioinformatics Resource, NCI, NIH, Bethesda, Maryland, United States of America, 4 Department of Comparative Medicine, Pennsylvania State University College of Medicine, Hershey, Pennsylvania, United States of America

◉ These authors contributed equally to this work.
¤ Current Address: Department of Immunology and Microbiology, School of Basic Medical Sciences, Wenzhou Medical University, Wenzhou, Zhejiang, China
* ndc1@psu.edu (NDC); zhengt@exchange.nih.gov (Z-MZ)

**Data Availability Statement:** All data underlying the findings described in this publication are freely available to other researchers. The RNA-seq data

## Abstract

Cottontail rabbit papillomavirus (CRPV), the first papillomavirus associated with tumor development, has been used as a powerful model to study papillomavirus pathogenesis for more than 90 years. However, lack of a comprehensive analysis of the CRPV transcriptome has impeded the understanding of CRPV biology and molecular pathogenesis. Here, we report the construction of a complete CRPV transcription map from Hershey CRPV-induced skin tumor tissues. By using RNA-seq in combination with long-reads PacBio Iso-seq, 5′ and 3′ RACE, primer-walking RT-PCR, Northern blot, and RNA *in situ* hybridization, we demonstrated that the CRPV genome transcribes its early and late RNA transcripts unidirectionally from at least five distinct major promoters (P) and polyadenylates its transcripts at two major polyadenylation (pA) sites. The viral early transcripts are primarily transcribed from three "early" promoters, $P_{90}$, $P_{156}$, and $P_{907}$ and polyadenylated at nt 4368 by using an early polyadenylation signal (PAS) at nt 4351. Like other low-risk human papillomaviruses and animal papillomaviruses, CRPV E6 and E7 transcripts are transcribed from three separate early promoters. Transcripts from two "late" promoters, $P_{7525}$, and $P_{1225}$, utilize either an early PAS for E1^E4 or a late PAS at 7399 for L2 and L1 RNA polyadenylation at nt 7415 to express capsid L2 and L1 proteins respectively. By using the mapped four 5′ splice sites and three 3′ splice sites, CRPV RNA transcripts undergo extensive alternative splicing to produce more than 33 viral RNA isoforms for production of at least 12 viral proteins, some of which without codon optimization are expressible in rabbit RK13 and human HEK293T cells. The constructed full CRPV transcription map in this study for the first time will enhance our understanding of the structures and expressions of CRPV genes and their contribution to molecular pathogenesis and tumorigenesis.

discussed in this publication have been deposited in NCBI's Gene Expression Omnibus are with an accessible number GSE124211.

**Funding:** This study was supported by Intramural Research Program of NCI/NIH [1ZIASC010357 to Z-MZ] and by the NCI grant (R01CA47622 to NDC). PJ was supported by the China Scholarship Council for 1-year study at NIH (CSC No. 201708330003). The funders had no role in study design, data collection and analysis, decision to publish, or preparation of the manuscript.

**Competing interests:** The authors have declared that no competing interests exist.

## Author summary

Papillomavirus infections induce benign warts and cancers. Naturally occurring infections in animal models have played a pivotal role in studying human papillomavirus pathogenesis. The cottontail rabbit papillomavirus (CRPV), the first reported papillomavirus, has been used widely for understanding the pathogenesis of papillomavirus-associated diseases and cancer. The current study uses the state-of-the-art RNA-seq and PacBio Iso-seq in combination with 5′ and 3′ RACE, primer-walking RT-PCR, Northern blot, and RNA *in situ* hybridization to determine genome-wide transcription and RNA structures of individual CRPV transcripts in Hershey CRPV-induced rabbit skin wart tissues. As a result, we have constructed the full transcription map from CRPV-induced rabbit skin warts and further determined the relative abundance and spatial localization of identified viral transcripts in these warts. The CRPV genome uses three early and two late viral promoters, four splice donor and three splice acceptor sites, and two viral PAS to express at least 33 RNA isoforms and twelve viral proteins. Most of their open reading frames were cloned for ectopic expression in this study. The constructed CRPV transcription map in this report will provide an important resource to advance our understanding of papillomavirus biology and pathogenesis.

## Introduction

The cottontail rabbit papillomavirus (CRPV) or Shope papillomavirus (SPV) was discovered by Dr. Richard Shope in 1933 as the first tumorigenic DNA virus [1–6]. Although the natural host of CRPV is the cottontail rabbits, domestic rabbits are susceptible to CRPV infection and skin cancer induction at a given time [7–9]. The CRPV/rabbit model has been widely used to study mechanisms of viral-host interactions, papillomavirus pathogenesis, and transmission ever since [5,10,11] and served as a surrogate preclinical model for high-risk human papillomavirus (HPV) infections [12]. Like high-risk HPV- and mouse papillomavirus (MmuPV1)-induced lesions [13–15], both episomal and integrated viral DNAs were found in CRPV-induced carcinomas in domestic rabbits, and high methylation of viral DNA was found in correlation with malignancies [16–21]. The development of CRPV cancers has been attributed to the malignant transformation of benign papilloma cells [22]. The incidence in development of cancers from papillomas in the domestic rabbits is threefold higher than those in the cottontail rabbits [23,24]. CRPV transcripts are detectable in benign and malignant rabbit papillomas [25–28].

Viral transcriptional and translational patterns were reported during the malignant progression of CRPV-induced papillomas in domestic rabbits [29–32]. Further studies identified that early genes E6, E7, E1, and E2 were essential for CRPV infection and papilloma formation [28,33–42]. With the development of new techniques for reproducible viral DNA infections [11,43], the CRPV/rabbit model can be used to test the outcome of viral gene function [44] and the impact of codon modifications in papillomavirus genome expression [45].

CRPV genome is a double-stranded, circular DNA genome in size of 7868 nucleotides for the prototype Shope strain [46] and 7871 nucleotides for the Hershey strain [47]. When compared with the Shope CRPV genome (S1 Fig), the Hershey CRPV genome has nine nucleotide insertions: G insertion after nt 3515, TTT insertions after nt 4342, CA insertions after nt 4371, C insertion after nt 5345, T insertion after nt 6890 and A insertion after 7613. It has also six nucleotide deletions: A deletion after nt 3532, three nucleotide deletion of ACA after nt 4361,

G deletion after nt 5324, and T deletion after nt 6928 (S1 Fig). The net gain in these insertions and deletions makes the Hershey CRPV genome three nucleotides longer than the Shope CRPV. The CRPV genome has been predicted to encode eleven proteins: long E6 (LE6), short E6 (SE6), E10 (previously called E8), E7, E1, E2, E1^E4, E8^E2, E5, L1, and L2 [6]. Although neither the LE6 nor SE6 protein is detectable during CRPV infection, both LE6 and SE6 by ectopic expression have transformation activity in NIH 3T3 cells. They do not interact with p53, E6AP, or E6BP [48, 49], but bind to hDlg/SAP97, another tumor suppressor [50], and histone acetyltransferase p300 from HEK293 cells [51] and tagged human MAML1 [52]. Recent studies showed that the CRPV E6 gene, especially the carboxyl-terminal of the regressive strain, plays a critical role in tumor regression in domestic rabbits [44, 53]. CRPV E6 and E7 mRNAs are highly expressed in malignant tumor tissues [32, 37, 48] and could be potential therapeutic targets for developing E6- and E7-based siRNAs or vaccines as being explored for HPV therapeutics and preventions [54–62]. ORF E10 (formerly E8) is colinear with ORF E6 in a different frame, which encodes 50-aa protein related to HPV E5 and has shown transformation activity in C127 cells in the presence of platelet-derived growth hormone (PDGF) [63] but not in NIH 3T3 cells [44,48]. E10, which is associated with ZnT-1 [64], is important in tumor outgrowth [63–66]. CRPV E5 was found to be dispensable for papilloma formation in domestic rabbits [42,44]. CRPV E1^E4 (or E4) is a late gene that is not required for tumor growth but plays a role in the productive viral life cycle [44, 67, 68]. CRPV E1 and E2 are highly homologous to E1 and E2 from other HPVs and play roles in viral DNA replication and transcriptional regulation [69]. L1 and L2 form CRPV capsid and are main viral immunogens [70–72].

To date, a partial transcription map of the CRPV early region has been reported [25,28,73]. However, the lack of a full CRPV transcription map has impeded the understanding of CRPV gene structure and expression, thereby, CRPV biology and molecular pathogenesis. The full transcription maps of HPV16, HPV18, HPV31, bovine papillomavirus type 1 (BPV1), and MmuPV1 (https://pave.niaid.nih.gov/explore/transcript_maps)[74–80] have been widely used for our understanding of gene structure and expression of individual papillomaviruses at different stages of virus infection, virus and host interactions, and molecular pathogenesis. In this report, we have constructed for the first time a full CRPV transcription map from Hershey CRPV-induced skin tumor tissues. The constructed CRPV transcription map and many novel findings, including viral promoters, novel splice sites, and usage of polyadenylation signals and cleavage sites, expression, and detection of individual promoter activities in the tissues, will provide an important resource to guide our understanding of CRPV gene expression during CRPV infection and oncogenesis.

## Results

### CRPV infection and genome expression profile in CRPV-induced wart tissues

To investigate CRPV transcriptome, the pre-wounded skin sites on the back of four male and four female New Zealand White (NZW) rabbits were experimentally infected by a serially diluted Hershey CRPV stock (from $10^{-2}$ to $10^{-5}$ of $2.75 \times 10^9$ virus genome equivalent) (Fig 1A). This infection protocol at a virus inoculum at $2.75 \times 10^7$ ($10^{-2}$), $2.75 \times 10^6$ ($10^{-3}$), or even at $2.75 \times 10^5$ ($10^{-4}$) led to papilloma (wart) formation in three weeks post-infection, as expected [12]. Significantly larger tumors were found in three higher doses ($1\times10^{-2}$ to $1\times10^{-4}$) compared to those of the lowest dose $10^{-5}$ and PBS control (Fig 1B, **p<0.01, *p<0.05, one-way ANOVA). Papillomas exhibited fibrillary projections accompanied by hyperkeratosis and were exophytic in morphology; sizing correlated positively with the virus load at week eight post-infection (S2A Fig). The tumors have a similar histopathology (H&E, S2B and S2C Fig)

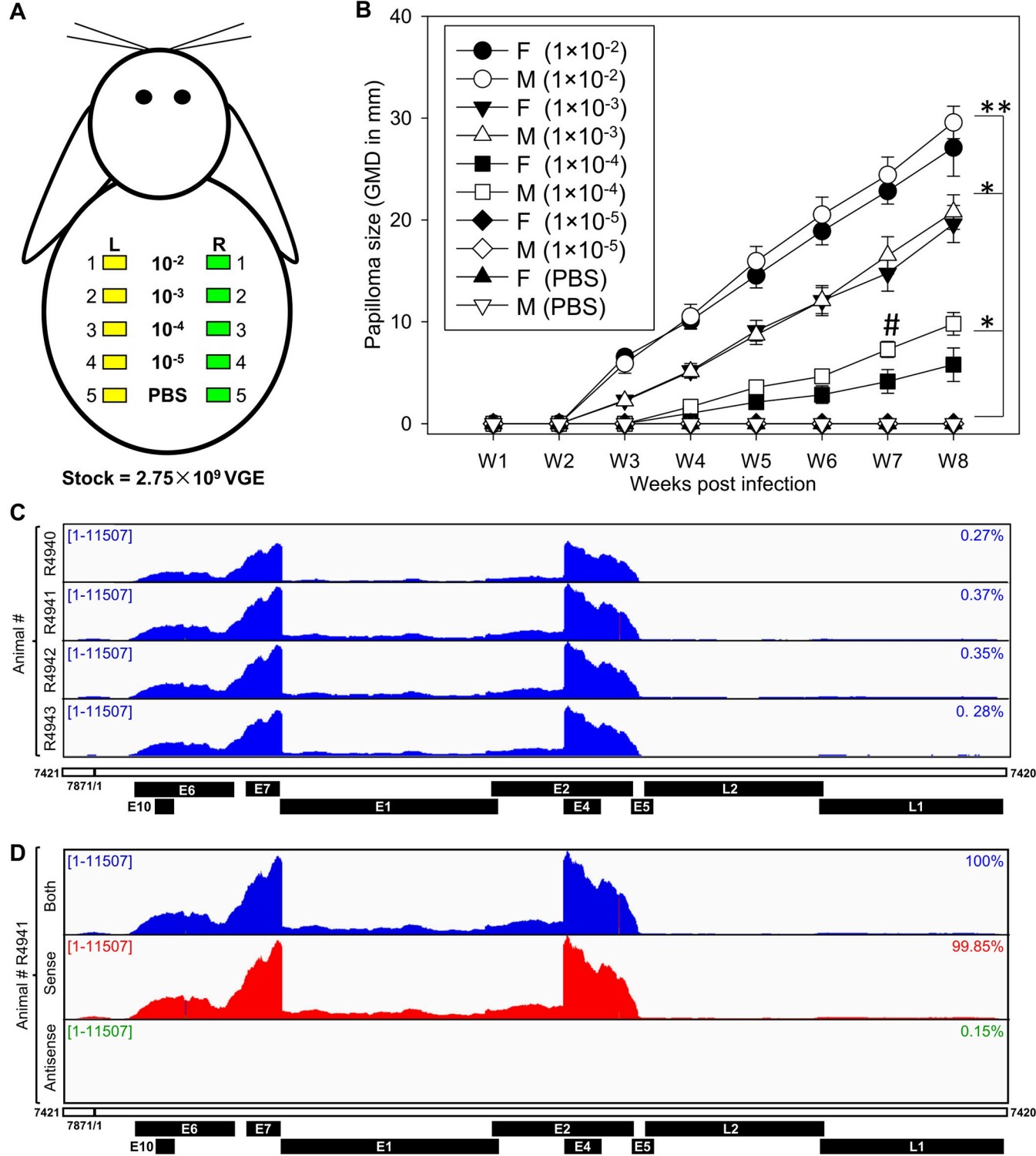

**Fig 1. RNA-seq analysis of viral gene expression in the CRPV-induced wart tissues.** (A) A diagram of the experimental infection of outbred New Zealand White (NZW) rabbits with CRPV Hershey strain. Four male and female rabbits were inoculated on the pre-wounded sites on the back with a serially diluted virus stock containing $2.75 \times 10^9$ of virus genome equivalent (VGE) in PBS. The same dose was applied on both the left (L, yellow box) and right (R, green box) back skin sites. Skin sites with PBS inoculation were used as negative controls. (B) The growth curves of tumors developed from different doses of infectious CRPV DNA (1:10 dilutions of the stock at $2.75 \times 10^9$ viral DNA equivalents). Most infected sites of the doses higher than the dilution of $1 \times 10^{-4}$ developed tumors at week eight post-infection. Significantly larger tumors, determined by geometric mean diameter (GMD) of height × width × length in mm, were

found in higher doses when compared with $1 \times 10^{-5}$ and PBS control groups (**p<0.01, *p<0.05, one way ANOVA). Comparable tumor size was found between males and females at all time points except week 7 (W7) post-infection (# p<0.05, unpaired Student t-test). The total RNA from the wart tissues at eight weeks post-infection was extracted separately from four female rabbits for RNA-seq analysis. (C) The distribution of RNA-seq reads mapped to the Hershey CRPV genome (GenBank Acc. No. JF303889.1) linearized at nt 7421 in four independent samples collected from four female rabbits. The CRPV reference genome with annotated potential ORFs is displayed in a linear form for better visualization of viral reads coverage by IGV. The numbers in the upper left corner represent the reads-count scale. The numbers on the right represent the percentage of CRPV-mapped reads from the total number of uniquely mapped reads (S1 Table). (D) The strand-specific expression from the CRPV genome. The viral-specific RNA-seq reads were mapped to both strands (shown in blue) and sense (red) or antisense (green) strand of the CRPV DNA genome (S1 Table). The reads distribution in one representative sample (animal #R4941) is shown at the same scale. On the right is the percentage of the total viral reads mapped to the individual genomic strands (S1 Table).

as papillomas for both male and female rabbits. The papillomatosis was characterized by papillary projections of hyperplastic squamous epithelium, hyperkeratosis, koilocytosis/koilocytotic atypia, and/or large numbers of lymphocytes in some samples (S2B–S2E Fig). Histopathological features of malignancy were not identified from these early-time lesions.

Eight weeks after infection, the warts were taken separately from four infected female rabbits for total RNA extraction and total ribo-minus RNA-seq. We obtained a total of ~60 million high-quality reads from each wart tissue sample (S1 Table). To elucidate RNA transcripts separately from the CRPV genome and the rabbit genome, we constructed a chimeric rabbit (*Oryctolagus cuniculus*)/CRPV (Hershey CRPV GenBank Acc. No. JF303889.1) genome. Alignment of high-quality RNA-seq reads from individual samples to the chimeric genome identified 125,539–212,313 reads uniquely mappable to the CRPV genome and 46,729,420–57,507,856 to the rabbit genome (S1 Table). To visualize the viral RNA reads-distribution along the CRPV genome, we artificially linearized the CRPV genome in the intergenic region downstream of viral L1 ORF with the linearized CRPV genome starting from nt 7421 and ending at nt 7420. The distribution of reads mapped to the linearized viral genome and visualized using the Integrative Genomics Viewer (IGV) program revealed the typical papillomavirus expression profile being highly consistent among four tumor samples, with the highest viral reads-coverage overlapping with E6, E7, and E4 ORF regions (Fig 1C). The sharp drop between E7 and E4 corresponds to a previously identified intron region spanning over the E1 and E2 ORFs [25,73]. The reads-coverage of other parts of the CRPV genome was significantly low. However, using a variable scale could clearly show other intron regions crossing over the CRPV genome that have not been reported during CRPV infection and replication (S3A Fig).

By mapping the viral reads separately to either plus or minus strand of the viral genome, we demonstrated virtually that all RNA-seq reads (99.85–99.88%) among all four tumor samples were sense transcripts (Figs 1D and S3B and S1 Table), indicating that, like all other papillomaviruses, CRPV expresses its genes unidirectionally in the sense orientation from the minus strand of the viral genome.

## Mapping of transcription start sites (TSS) of early and late CRPV transcripts

To map the Hershey CRPV promoters, we analyzed transcription start sites (TSS) of viral transcripts by 5′ RACE on total RNA isolated from the Hershey CRPV tumor tissues using a set of virus-specific antisense primers (Pr) from the viral genome based on RNA-seq reads-coverage (Fig 2A). The obtained 5′ RACE products were gel-purified, TOPO cloned, and sequenced by Sanger sequencing (Fig 2B and S2 Table). Alternatively, gel-purified 5′ RACE products were subjected to direct sequencing. We found that primer located at nt position 7812 (Pr7812) from the upstream regulatory region (URR) produced a single 5′ RACE product, with most of the colonies containing TSS at nt 7525 (Fig 2B, lane 1 and S2 Table). Other 5′ RACE products subjected to TOPO cloning and sequencing included: two major products obtained from

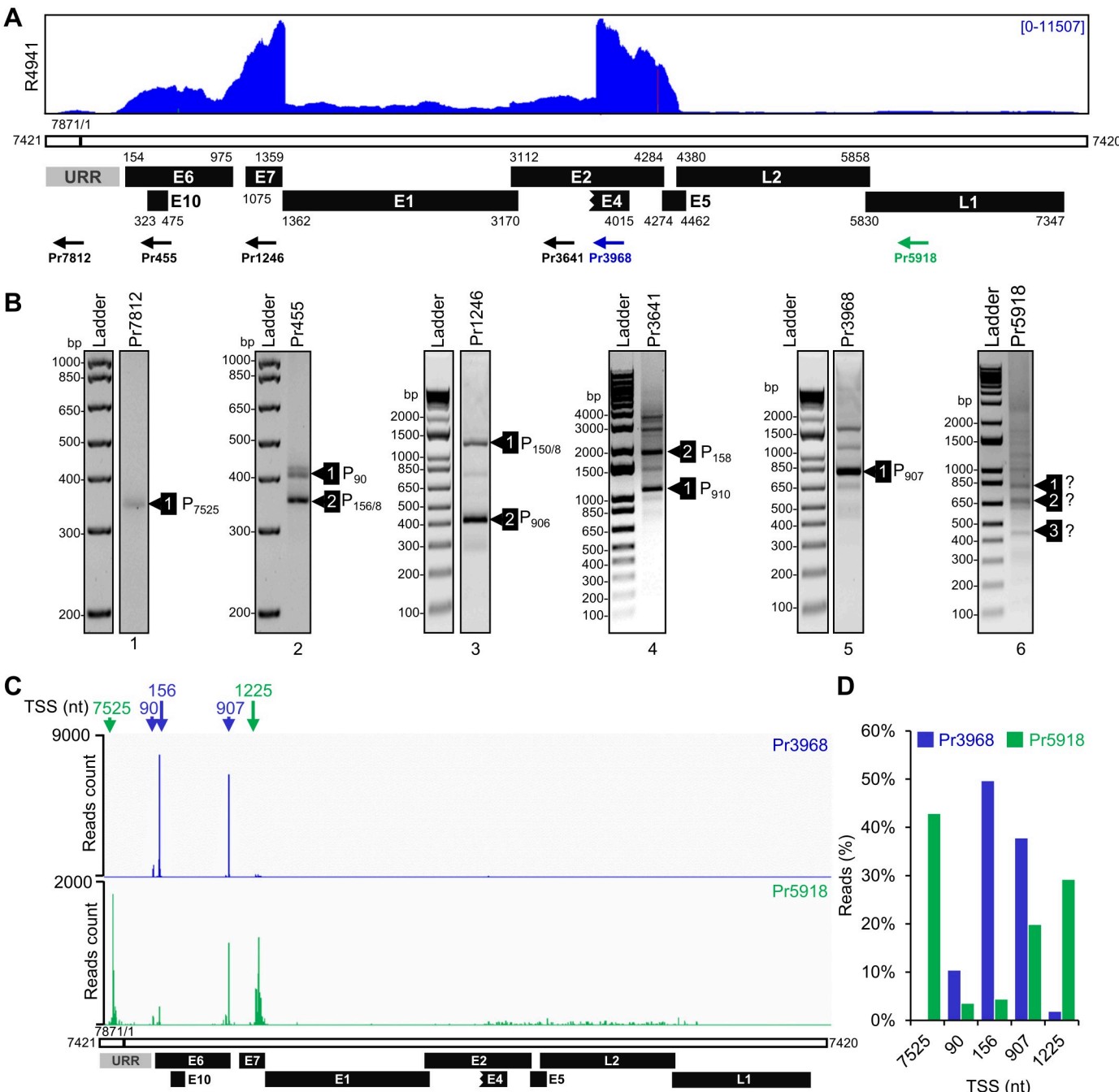

**Fig 2. Mapping of CRPV promoter transcription start sites in rabbit tumor tissues.** (A) A diagram depicting the distribution of RNA-seq reads from a representative sample along with the linearized CRPV genome and its major annotated viral ORFs (black boxes). Numbers indicate viral genomic positions. URR, upstream regulatory region (grey box). The arrows beneath the diagram represent the primers used in 5′ RACE. (B) Gel electrophoresis of the 5′ RACE products obtained by primers targeting different parts of the viral genome shown in (A). The identity of marked products was confirmed by Sanger sequencing after cloning into the pCR2.1-TOPO vector. P represents the most frequently detected transcriptional start sites (TSS) (S2 Table). (C) The frequency and distribution of viral TSS based on PacBio sequencing of the 5′ RACE products generated by an early region primer Pr3968 (blue) or a late region primer Pr5918 (green). The arrows below indicate the genome positions of TSS with the highest number of PacBio reads (S3 Table). (D) The usage of mapped viral early and late promoters in rabbit tumor tissues. The percentage of PacBio reads mapped to the individual promoter TSS (±5 nt) in early (blue) or late (green) transcripts. See S3 Table for details.

Pr455 in the E6 ORF having TSS mapped at nt 90 or nt 156/158 (Fig 2B, lane 2), two major products obtained from Pr1246 in the E7 ORF having TSS at nt 906 or nt 150/158 (Fig 2B, lane 3), and two major products from Pr3641 in the E2 ORF with a splice junction (SJ) from nt 1371 to nt 3065 but from two separate TSS at nt 910 and nt 158 (Fig 2B, lane 4). A major 5′ RACE product from Pr3968 in the E4 ORF exhibited a TSS at 907 and a splice junction of nt 1371^3714 by direct sequencing of the gel-purified products (Fig 2B, lane 5). All TSS mapped by 5′ RACE were consistent among CRPV tumor tissues from four tested rabbits (S4A Fig). Because of a minimal amount of L1 RNA in the tumor tissues (Fig 2A) and heterogenic 5′ RACE products weakly amplified by Pr5918 in the L1 ORF, we were unable to precisely map their TSS by direct sequencing of gel-purified products or by TA-cloning and sequencing (Fig 2B, lane 6 and S2 Table).

To further perform unbiased quantification frequency of transcription initiation from individual promoters during CRPV infection, the obtained 5′ RACE products from Pr3968 and Pr5918 were also subjected to single-molecular, real-time sequencing (SMRT) of long reads (up to 30 kb) using PacBio Iso-seq technology (Figs 2C and S5A). Alignment of the obtained PacBio long reads derived from Pr3968 5′ RACE products to the Hershey CRPV genome linearized at nt 7421 revealed several distinct peaks within URR, E6, and E7 regions, each corresponding to the individual viral promoters. The identified TSS closely corresponded to the TSS identified by TOPO cloning and sequencing (Fig 2B and 2C and S3 Table). More importantly, we were able to map each TSS from Pr5918-derived 5′ RACE products. Subsequently, we named the viral promoter (P) based on the nucleotide position of the most frequently mapped TSS. The identified viral promoters are: $P_{7525}$, $P_{90}$, $P_{156}$, $P_{907}$, and $P_{1225}$ (Fig 2C). Sequencing of 5′ RACE products obtained using Pr5918 from the L1 region in a separate SMRT revealed additional putative promoters with TSS at nt 4258 upstream of L2 ORF and nt 5765 within CRPV L2 ORF (S5A Fig), but more rigorous validation of these two putative TSS are needed in future studies.

To determine the relative usage of individual promoters during CRPV infection, we compared the number of associated PacBio reads (TSS -/+ 5 nt) from Pr3968 in the E4 region, detecting mainly the early transcripts with that from Pr5918 in the L1 region detecting only the late transcripts (Fig 2D and S3 Table). This comparative analysis showed, as expected, that the early transcripts (Pr3968) were transcribed exclusively from promoters $P_{156}$ (50%), $P_{907}$ (38%), and $P_{90}$ (~10%), whereas the late transcripts primarily originated from promoters $P_{7525}$ (~43%), $P_{1225}$ (~29%) and $P_{907}$ (~20%) (Figs 2D and S5 and S3 Table). Notably, the identified Hershey CRPV promoters are close to the genomic positions of reported Shope CRPV promoters mapped at nt 95 (P1), nt 165 (P2), and nt 908 (P3) [26, 28] or at nt 87, nt 157–159, nt 903–908, nt 970, and nt 975 [73] for transcription of viral early genes and at nt 7519–7523 (PL) for transcription of viral late genes [28]. The novel promoter $P_{1225}$, newly identified in CRPV E7 ORF region in this report, resembles MmuPV1 late promoter $P_{533}$ and HPV18 promoter $P_{812}$ [76, 80] as well as HPV31 and HPV16 promoters [81, 82]. Based on these data, we concluded that two viral early promoters, $P_{156}$ and $P_{907}$, along with a minor early promoter, $P_{90}$, are responsible for the expression of viral early transcripts, while the promoters $P_{7525}$ and $P_{1225}$ are two major late promoters for the expression of viral late RNAs. Interestingly, the promoter $P_{907}$, a viral early promoter mainly for E7 expression, remained active at the late stage of CRPV infection (Fig 2C and 2D).

In agreement with conserved TSS starting often at a purine in eukaryotes [83], all mapped CRPV TSS start at an A ($Pr_{7525}$) or G ($P_{90}$, $P_{156}$, $P_{907}$, and $P_{1225}$). The sequence analyses of the upstream region to each mapped early TSS revealed a canonical TATA box (a eukaryotic core promoter motif) ~30-bp upstream of the mapped TSS (S6 Fig). In contrast, both viral late promoters, $P_{7525}$ and $P_{1225}$ lack an optimal consensus TATA box in their immediate upstream

region. Instead, the promoter $P_{7525}$ has a TATA-like TATT at 98-bp and a TATA box at 129-bp far upstream of the mapped TSS, and the promoter $P_{1225}$ bears only a TATA-like TTTA motif at 61-bp and a TATA box at nt 197-bp, again, far upstream of the mapped TSS. These features of viral late promoters perhaps account for the observed heterogeneity of their transcription initiation, as seen in the late gene expression of MmuPV1, HPV18, and HPV31 [76,80,84].

## Mapping of CRPV polyadenylation cleavage sites for early and late transcripts

Viral transcripts of other papillomaviruses are terminated by host polyadenylation machinery using two viral polyadenylation signals (PAS), early and late PAS [74, 85]. Sequence analyses of the Hershey CRPV genome revealed four canonical PAS, AAUAAA, in its plus strand at nt 4351, 4470, 6193, 7399. To determine the Hershey CRPV early polyadenylation cleavage sites (CS), total RNA isolated from the Hershey CRPV-infected lesions was analyzed by 3′ RACE using a sense primer Pr3922 located within the E4 coding region (Fig 3A). The obtained ~550 bp product (Fig 3B, lane 1) was cloned, and 16 individual colonies were subjected to Sanger sequencing. We found that 10/16 sequenced bacterial colonies exhibited a product with a 3′ end mapping to nt 4368, 17 nt downstream of the canonical PAS at nt 4351 (Fig 3C), indicating that the most Hershey strain CRPV early transcripts are cleaved and polyadenylated at nt 4368 notably in the annotated E5 coding region. To detect the late transcript polyadenylation CS, a sense primer Pr7221 located in the L1 coding region was used for the 3′ RACE, resulting in a single ~300 bp RACE product (Fig 3B, lane 2). TA cloning of the gel-purified RACE products and sequencing of 14 bacterial colonies identified nt 7415 as the most frequent polyadenylation CS, 16 nt downstream of a canonical PAS at nt 7399 for the Hershey CRPV late transcripts (Fig 3C). The mapped cleavage sites of CRPV early and late transcripts were confirmed in all CRPV tumor tissues of the tested four individual rabbits (S4B Fig). Together, we have precisely mapped in this study the polyadenylation CS downstream of each PAS in the Hershey CRPV genome for the expression of viral early and late transcripts. In the Shope CRPV (NC_001541.1), an early PAS at nt 4348 was found to guide for RNA polyadenylation at nt 4368 [73], and a late PAS at nt 7397 was presumably being used, but not experimentally confirmed yet, for the expression of the viral late genes.

## Alternative splicing of CRPV transcripts

In addition to the usage of the alternative promoters and polyadenylation sites mapped above, Hershey CRPV also applies alternative RNA splicing to diversify its transcriptome for viral gene expression. To determine a genome-wide alternative RNA splicing of Hershey CRPV, we extracted viral splicing junctions (SJ) from RNA-seq reads using STAR aligner (S4 Table). In total, we identified 37 unique SJ, of which 36 are from canonical GU:AG intron splicing and only one from a noncanonical GC:AG intron splicing with one read from one sample (S4 Table). To determine the most prevalent splicing events, we combined the number of SJ reads from all four tissue samples and obtained a total of 60,267 viral SJ reads. We further identified the seven most prevalent SJ with more than 50 SJ reads in all four tissue samples (S4 Table). These seven SJ were generated by alternative RNA splicing from four splice donor sites (SD, 5′ splice site) at nt 7813, 1371, 1751, and 4079 and three splice acceptor sites (SA, 3′ splice site) at nt 3065, 3714, and 5830 (Fig 4A). The position and frequency of these SJ (labeled as 1–7) visualized by the Sashimi plot along with the linearized Hershey CRPV genome are shown in Fig 4A. It is obvious that the RNA splicing between the SD site at nt 1371 and the SA site at nt 3714 represents a predominant splicing event in the tumor tissues and counts 57,394 or ~95%

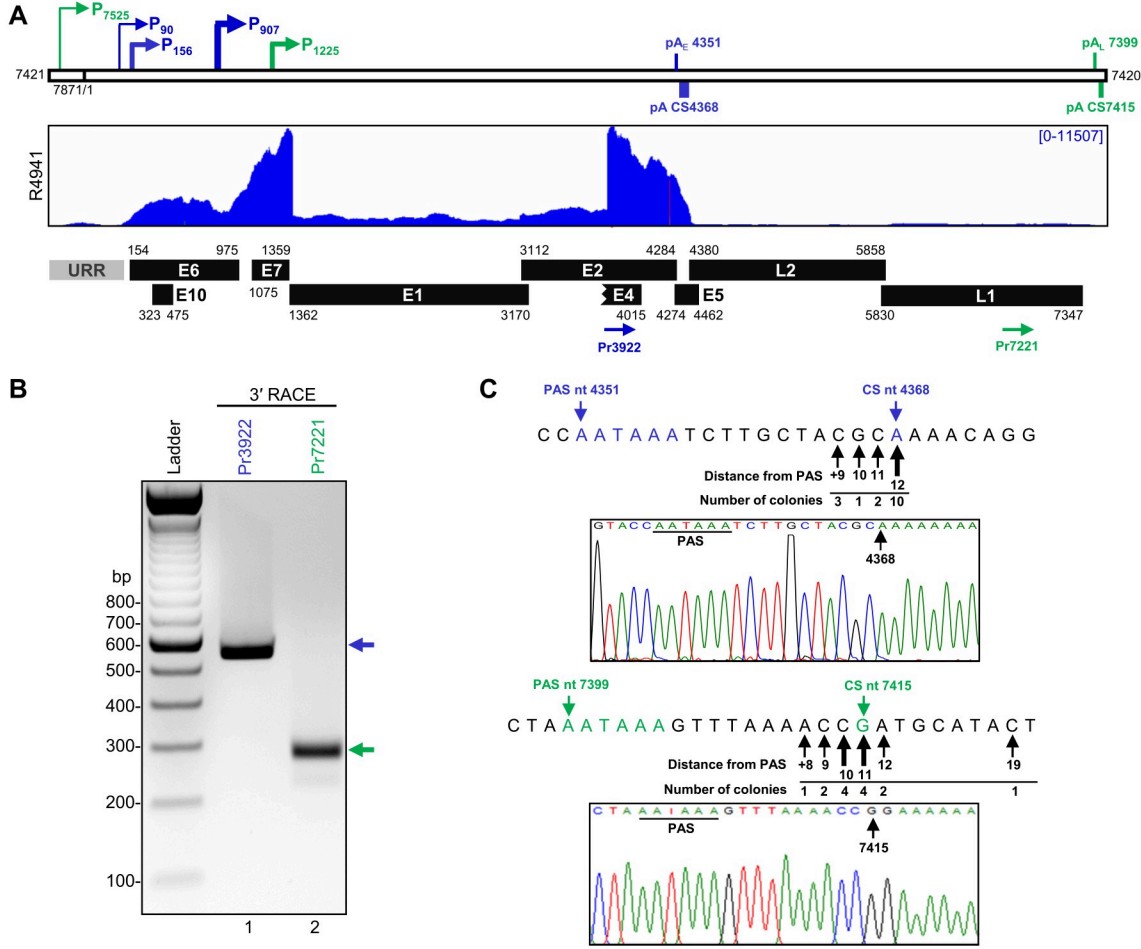

**Fig 3. Identification of polyadenylation cleavage sites for CRPV transcripts.** (A) Diagrams of the linearized CRPV genome with the positions of mapped viral promoter transcription start sites (colored arrows). The horizontal lines mark the position of predicted early (blue) and late (green) PASs (early pA$_E$ and late pA$_L$) and polyadenylation cleavage sites (pA CS) mapped by 3′ RACE using the primers depicted as arrows below. Under is a distribution of mapped viral RNA-seq reads obtained from one representative CRPV tumor tissue and CRPV annotated ORFs (black boxes) with their corresponding genomic positions. (B) The electrophoresis gel image showing the 3′ RACE products obtained using the indicated primers from total RNA extracted from CRPV tumor tissues. The individual bands were gel purified, cloned, and sequenced by Sanger sequencing. (C) The nucleotide sequence of the predicted CRPV polyadenylation signal (PAS) and the most prevalent pA CS identified by 3′ RACE in this study. The arrows below the sequence point to the mapped alternative CSs, the number of detected colonies, and their distance from PAS. The representative Sanger sequencing chromatographs show the use of the most prevalent pA CS followed by a polyA tail.

of all detected SJ reads, whereas the second most prevalent splicing event from nt 1371 to nt 3065 counts only 1765 SJ reads or ~2.9% of all SJ reads. All remaining RNA splicing events count only ~1.8% of detected SJ reads (Fig 4A).

To experimentally confirm SJ identified by the STAR aligner, we performed primer-walking RT-PCR using serial sets of primer pairs (blue arrows in Fig 4A). The sequencing of gel-purified RT-PCR products confirmed all seven major SJ identified in the Hershey CRPV genome: 7813^3714, 7813^5830, 1371^3065, 1371^3714, 1751^3065, 1751^3714, and 4079^5830 (Fig 4B). Among them, five splice sites (7813^3714, 1371^3065, 1371^3714, 1751^3065, 1751^3714) were reported by other labs [25,28,73]. We further verified the splice junction 4079^5830 and 7813^5830 for L1 expression in CRPV tumor tissues separately from

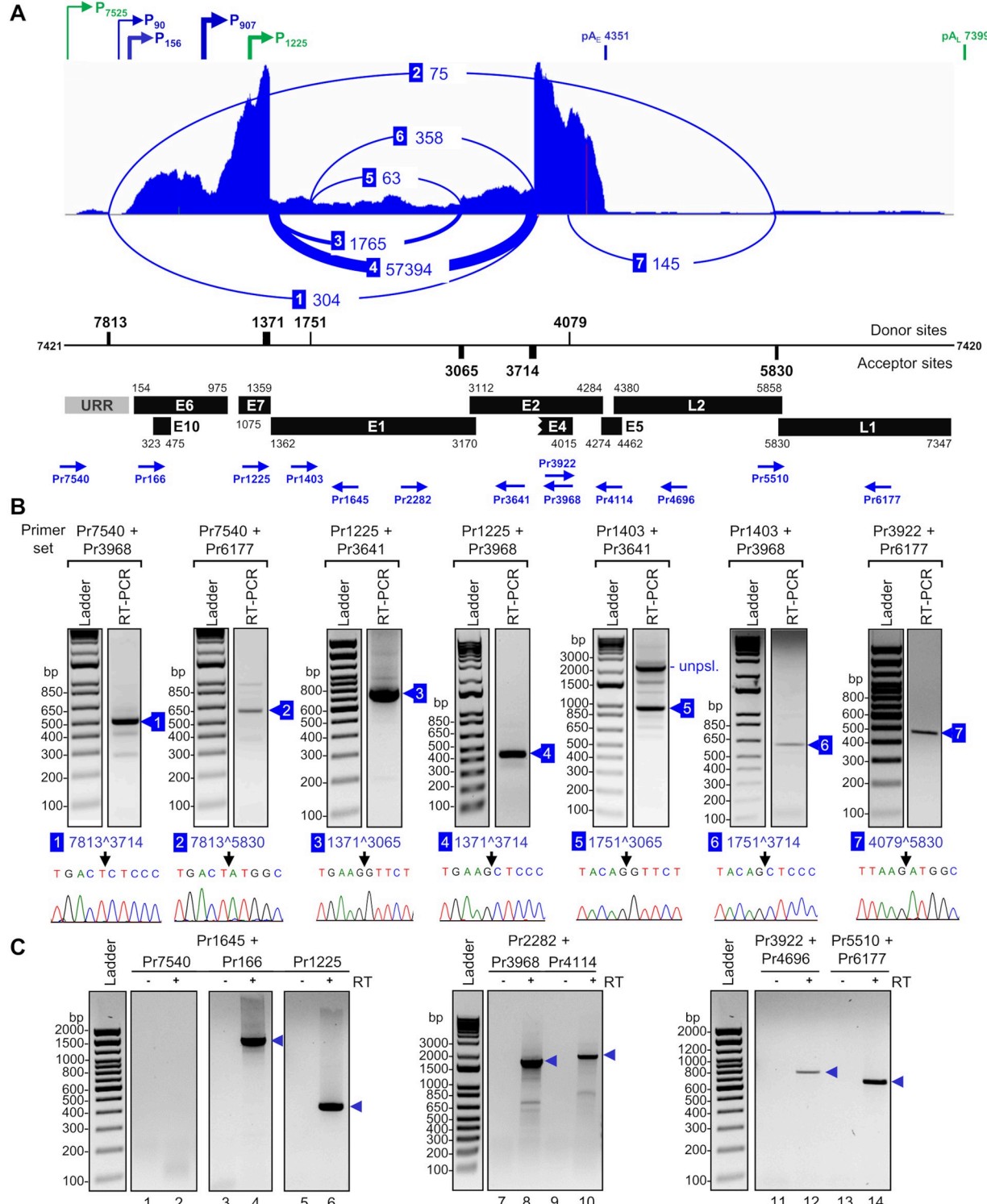

**Fig 4. CRPV RNA splicing profile in CRPV tumor tissues.** (A) Detected viral splice junctions (SJ) and intron retention, with the mapped early (blue) and late (green) promoters and PAS above the splicing profile. The distribution of RNA-seq reads from one representative CRPV tumor tissue mapped to the linearized CRPV genome with a Sashimi plot (numbered arches) to visualize the viral SJ positions. The arches' thickness corresponds to the number of detected SJ reads in all 4 samples (S4 Table). Only SJ supported with >50 reads are shown. Below the Sashimi plot are the nucleotide positions of the identified splice donor and acceptor sites for shown SJ together with viral annotated ORFs and their genomic positions. URR, upstream regulatory region. Blue arrows represent the primers used for validation of mapped SJ or intron retention by RT-PCR.

(B) Experimental validation of CRPV RNA splicing. Gel electrophoresis of RT-PCR products obtained using the primers shown in (A) on DNase-treated total RNA isolated from CRPV tumor tissues. The numbered blue boxes indicate the RT-PCR products selected for Sanger sequencing. The confirmed SJ positions are marked by arrows in the corresponding chromatographs beneath each gel picture. (C) Intron retention in CRPV transcripts in CRPV tumor tissues. The gel electrograph of RT-PCR products obtained using the indicated primer pairs shown in (A) on DNase-treated total RNA from CRPV-induced wart tissues. Reactions without reverse transcriptase (- RT) were used as negative controls. The individual bands in the predicted size were gel-purified and confirmed by Sanger sequencing.

four individual rabbits (S4C Fig). Using a threshold of 10 SJ reads as a cutoff, we also identified 5 additional SJ at nt 7573^3714, 868^3714, 1478^3714, 3231^3714, and 1371^5830 in the Hershey CRPV expression (S7A Fig and S4 Table). Despite their low prevalence we confirmed some experimentally by using RT-PCR (S7B Fig).

## Viral intron retention during CRPV genome expression

Generally, a papillomavirus genome contains two (low-risk HPV and animal papillomaviruses) or three (high-risk HPV) common introns with alternative splice sites for alternative RNA splicing in the expression of viral genes. Intron retention is one of the common alternative RNA splicing events during RNA processing [86] and is essential for the expression of several papillomavirus proteins, including E1, E2, E6, and L2, of which ORFs span the intron region. Splicing of these introns will destroy the integrity of these ORFs and thus abolish their expression of full-length proteins [62,74,87–89]. Although the distribution of RNA-seq reads shows that most viral early and late transcripts in the CRPV tumor tissues are spliced, we also observed a low but continuous coverage within the intron regions of the Hershey CRPV genome (S3A Fig), indicating a low percentage of intron retention in the expression of CRPV genome. To confirm the expression of these intron regions, we performed a primer-walking RT-PCR using several sets of primer pairs (blue arrows in Fig 4A) to detect the retained introns in DNase-treated total RNA extracted from the Hershey CRPV-induced tumor tissues. The reaction without RT was used as a control to ensure the amplified product originated from RNA and not from contaminating viral DNA. As shown in Fig 4C, the expression of E1 was confirmed by an antisense primer Pr1645 located in the E1 coding region in combination with forward primers Pr166 or Pr1225 (lanes 4 and 6), but not Pr7540 (lane 2). Data indicate that E1 RNA could be transcribed from both early promoters $P_{90}$, $P_{156}$, and $P_{907}$ or the late promoter $P_{1225}$, but not from late $P_{7525}$. Similarly, retention of the intron spanning E1 and E2 ORF regions was confirmed by RT-PCR using a forward primer Pr2282 in combination with an antisense primer Pr3968 or Pr4114 (Fig 4C, lanes 8 and 10). As the primer Pr4114 is downstream of the nt 4079 splice donor site, the reduced band density from Pr4114 (lane 10) over the Pr3968 (lane 8) in the RT-PCR could reflect the amplification of only the RNA transcripts being skipped from the nt 4079 splicing (Fig 4C). Retention of the intron spanning the L2 ORF region was confirmed by RT-PCR using two sets of primers, a forward primer Pr3922 in the E2/E4 ORF region in combination with an antisense primer Pr4696 in the L2 ORF region (Fig 4C, lane 12) and a forward primer Pr5510 in the L2 ORF in combination with an antisense primer Pr6177 in the L1 ORF region (Fig 4C, lane 14). More RNA amplification by the primer set Pr5510 plus Pr6177 (lane 14) than the primer set Pr3922 plus Pr4696 (lane 12) could result from the annealing and priming efficiency of individual primers used in the RT-PCR. In all cases, the RT-PCR led to amplification of the L2 RNA with an expected size only in the presence of reverse transcriptase. Their identity was further confirmed by Sanger sequencing after gel purification.

## Construction of a full CRPV transcription map

The mapped TSS, polyadenylation cleavage sites, and major RNA splice sites spanning over two major introns in the genome allowed us in this study to construct a full CRPV transcription map (Fig 5) in correlation to the RNA-seq profile (Fig 4A) in Hershey CRPV tumor tissues. Based on this CRPV transcription map in Fig 5, the Hershey CRPV genome might express 12 viral proteins, presumably from three translational frames (Fig 5A) and 33 different isoforms of transcripts (Fig 5C and S5 Table). Hershey CRPV early transcripts originated from the promoter $P_{90}$ are likely for LE6, $P_{156}$ for E10 or SE6, and $P_{907}$ for E7 expression. Most early transcripts are polyadenylated at an early polyadenylation CS at nt 4368 using a viral early poly A signal $pA_E$ at nt 4351 in the annotated E5 ORF region (Fig 3C) and thus, this polyadenylation will disrupt the integrity of the E5 ORF. Occasionally, some transcripts originated from the early promoters (transcripts I, J, P, R, X, and Y) may be polyadenylated by using a late $pA_L$ signal at nt 7399 by transcription reading through the early $pA_E$ signal (Fig 5C). Most early transcripts are spliced between 1371^3714, spanning the E1 and E2 coding regions and encode LE6, SE6, E10, and E7. E2 expression is presumably mediated by nt 1371^3065 splicing of early transcripts E, L, T, and AA, but L and T could be relatively more abundant than other two species, while E1 could be expressed only by retention of the E1 intron (transcripts H, O, W, and AD). Splicing of additional weak splice sites from nt 1751 within the E1 coding region to a splice site at nt 3065 or 3714 in the major early transcripts derived from the promoter $P_{156}$ or $P_{907}$, may lead to the expression of additional proteins, including E1^M1 or E1^M2 and E8^E2 (transcripts F, G, M, N, U, V, AB, and AC).

The late L1 and L2 transcripts, although being counted only about ~2.2% of the total viral RNA. The late transcripts are transcribed mainly from two late promoters, $P_{7525}$ and $P_{1225}$, and polyadenylated either at an early or late polyadenylation CS (Fig 5C). The promoter $P_{7525}$, located in URR, drives primarily the expression of late proteins L1 and L2. Even though the $P_{7525}$ transcripts span the entire early region, the early ORFs are removed by alternative RNA splicing from nt 7813 to nt 5830 (transcript A) or first to nt 3714 and then from nt 4079 to nt 5830 (transcript B) for expression of L1. To express L2, the late promoter $P_{7525}$ transcripts need to be spliced from nt 7813 to nt 3714, read through the early $pA_E$ signal (transcript C), and retain the L2 intron from RNA splicing before polyadenylating at a late $pA_L$ at nt 7399. Most transcripts from the late promoter $P_{1225}$ are polyadenylated from the early $pA_E$ cleavage site and spliced from nt 1371 to nt 3714 for expression of late E1^E4 protein (transcript Z). Some of them are polyadenylated at the late $pA_L$ cleavage site by double or single RNA splicing events to encode L1 and L2 (transcripts AE and AF). Moreover, two additional putative promoters, $P_{4258}$ and $P_{5765}$, were identified by analysis of 5′ RACE PacBio reads generated by an antisense primer in the L1 ORF but not verified by other means. We propose the putative $P_{4258}$ transcripts for L2 (transcript AG) and the $P_{5765}$ transcripts for L1 expression (transcript AH).

In summary, the majority of CRPV transcripts are identified to be polycistronic and alternatively spliced. They are generated by combinatorial use of alternative promoters, alternative RNA splicing, and alternative polyadenylation sites. Because polyadenylation of the early transcripts disrupts the E5 ORF region, the constructed Hershey CRPV transcription map does not have an intact E5 ORF (Fig 5). The transcript composition and predicted sizes are summarized in the S5 Table.

## Northern blot detection of viral transcripts from Hershey CRPV-induced tumor tissues

Next, we wished to validate the predicted CRPV transcriptome in Hershey CRPV-induced tumor tissues. We performed a Northern blot analysis of total RNA isolated from the wart

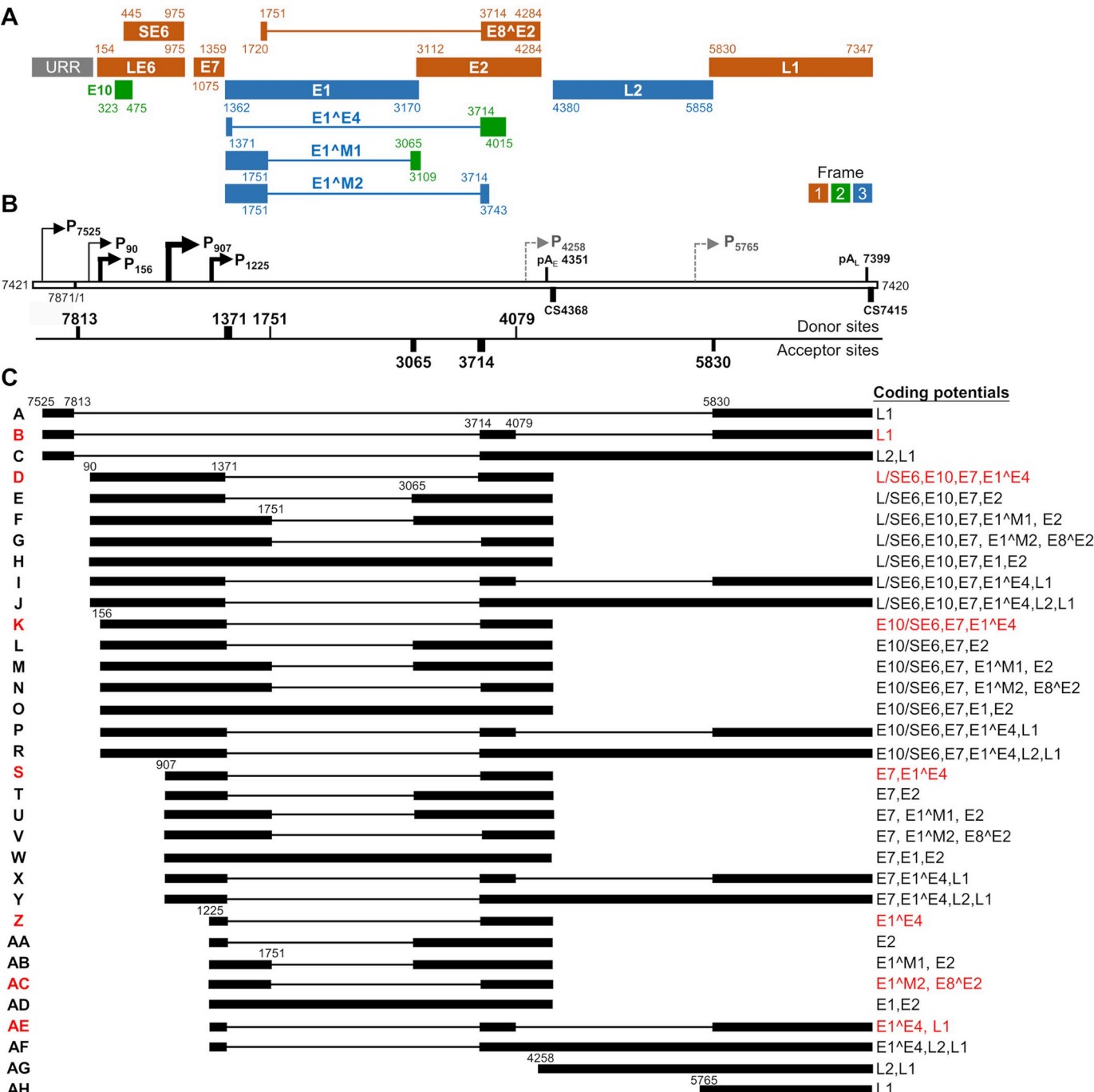

**Fig 5. Constructed transcription map of major CRPV transcripts from Hershey CRPV tumor tissues.** (A) The predicted viral open reading frames (ORFs) with nucleotide coordinates are shown below. The colors marked 1–3 represent the frames used for translating individual viral proteins, with E6 ORF being set as frame 1. The URR stands for the upstream regulatory region. (B) Nucleotide positions of the mapped viral promoters (P), polyadenylation (pA) signals and cleavage sites (CS), and viral splicing donor and acceptor sites (see Figs 2–4 for details). Dashed arrows, putative promoters. (C) Structure of the most prevalent viral transcripts, named from A-AH, with thick lines representing the exons and thin lines for the introns (see S5 Table for details), with the coding potentials of individual transcripts displayed on the right side. The transcripts in red color indicate relatively abundant viral transcripts commonly detectable in Hershey CRPV tumor tissues.

tissues using a series of $^{32}$P-labeled antisense oligos (black arrows in Fig 6A) designed according to the mapped promoters. The first set of oligo probes, Probe 120, Probe 455 and Probe 972 were used to detect viral early transcripts, respectively, from three viral early promoters, $P_{90}$, $P_{156}$, and $P_{907}$. The second set of oligo probes, Probe 1246 and Probe 3968, were used for the detection of both early and late transcripts, including those transcribed from two viral late promoters $P_{1225}$ and $P_{7525}$. As shown in Figs 6B and 5C and S5 Table, Probe 120 immediately downstream of $P_{90}$ detected three distinct products, the most abundant ~1.9-kb transcript D with single nt 1371^3714 splicing and less ~2.3-kb transcript G with nt 1751^3714 splicing, and ~4.3-kb transcript H without RNA splicing (lane 1). These $P_{90}$ transcripts presumably encode LE6, SE6, E7, and E10. Probe 455 immediately downstream of the promoter $P_{156}$ detected three $P_{156}$-transcribed RNA products with or without alternative RNA splicing, ~1.9-kb transcript K, ~2.5-kb transcript L, and ~4.2-kb transcript O in addition to $P_{90}$ transcripts D, E, G and H (lane 3). Due to the small difference in size, these $P_{156}$ transcripts are indistinguishable from the $P_{90}$ transcripts, but their presence is confirmed by a significant increase in signal intensity from each product. Probe 972 downstream of the promoter $P_{907}$ detected all transcripts from both promoters of $P_{90}$ and $P_{156}$, but also detected an additional ~1.1-kb transcript S originated from $P_{907}$ with nt 1371^3714 splicing (lane 5) to encode mainly viral E7 protein, although detecting a ~1.8-kb transcript T for E7 and E2, ~2.1-kb transcript U, ~1.5 transcript V, and ~3.4-kb transcript W might be possible. The Probe P1246 downstream of the promoters $P_{90}$, $P_{156}$, $P_{907}$, and $P_{1225}$ could detect, in principle, all transcripts derived from these four promoters including transcripts Z to AF, originated from the late promoter $P_{1225}$. A ~0.8-kb late transcript Z encoding E1^E4 was detectable only from the promoter $P_{1225}$ (lane 7). Finally, Probe 3968, downstream of the major nt 3714 splice acceptor site but upstream of the nt 4079 splice donor site, detected all transcripts from all mapped viral promoters, including both early and some late transcripts, most importantly the promoter $P_{7525}$-derived transcripts B (~2.2-kb) and C (~4-kb) for production of L1 and L2 proteins. Probe 3968 also gave a better detection of transcripts Z (~0.8-kb), AD (~3.1-kb), and AF (~3.8-kb), despite the transcripts spanning L1 and L2 regions were normally below a detectable level due to very low expression levels (lane 9). In summary, our Northern blot analysis showed that viral early transcripts originated from promoters $P_{90}$, $P_{156}$, and $P_{907}$, spliced from nt 1371 to nt 3714, and polyadenylated at an early PAS were the most abundant viral transcripts in the Hershey CRPV wart tissues. Although the bands assigned for our Northern blots were by presumption from the transcriptomic and RT-PCR data, their abundance is in accordance with the observed RNA-seq coverage and TSS analysis.

## Differentiation-dependent expression of CRPV transcripts determined by RNAscope: RNA *in situ* hybridization (RNA-ISH)

Differentiation-dependent viral gene expression is a hallmark of papillomavirus gene regulation [85, 90]. To investigate the expression of identified CRPV transcripts *in situ* within infected wart tissues as reported [32], we performed RNA-ISH using several RNAscope probes (Fig 6A, brown color) targeting the CRPV genome regions of E6, E2, E4, and L1. The probe targeting host keratin 10 (Krt10), a differential marker of spinous keratinocyte differentiation, was used as a positive control. Our previous study of MmuPV1 RNA-ISH using an antisense probe showed the detection of both viral RNA and one strand of double-stranded viral DNA [80]. To distinguish RNA-ISH signal specifically originated from viral RNA but not viral DNA, we performed the RNA-ISH analysis in parallel using serial tissue sections of the CRPV wart tissues with or without DNase I treatment as in our MmuPV1 report [80].

**A**

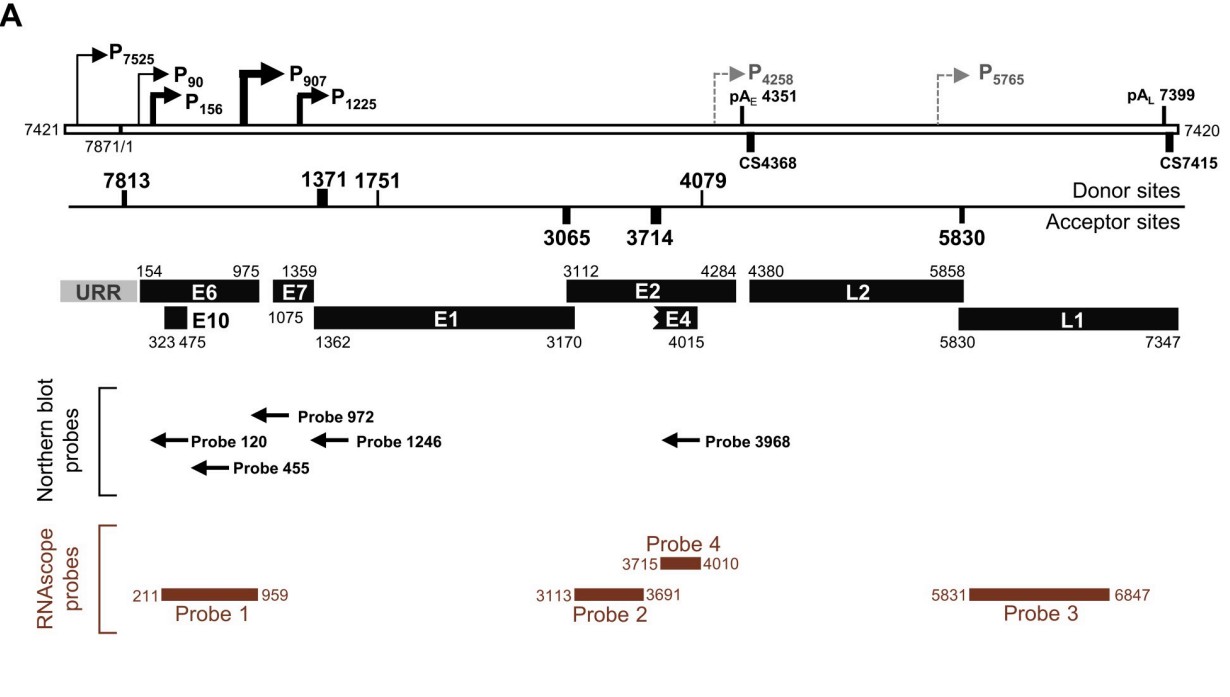

**B**

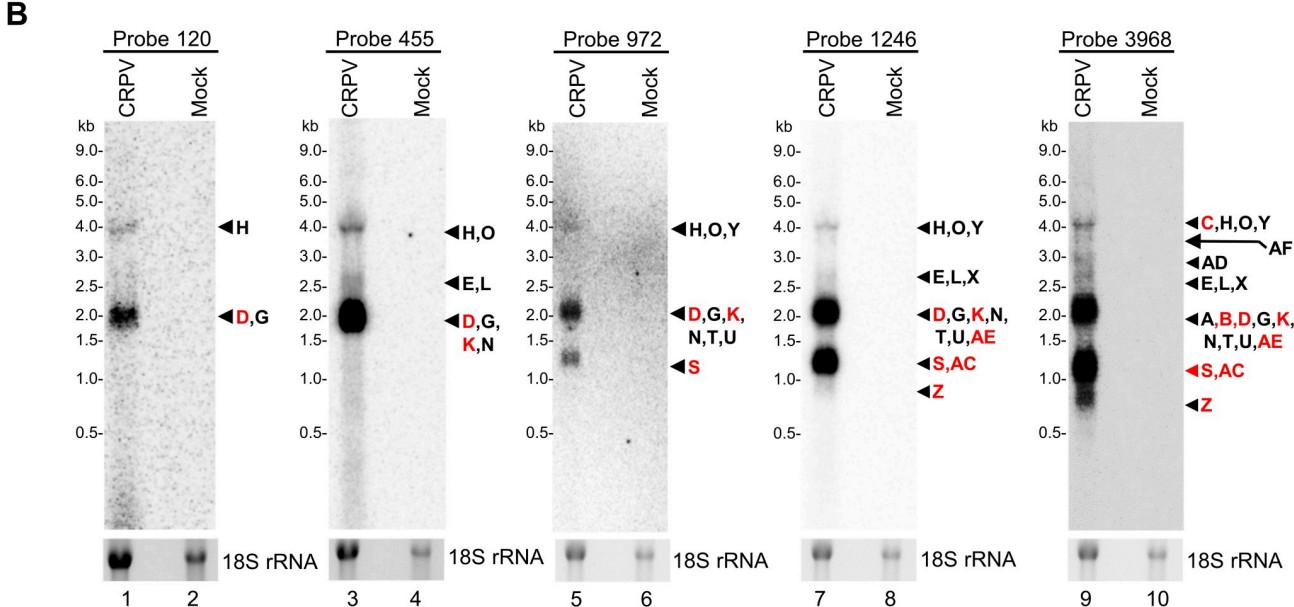

**Fig 6. Northern blot detection of viral transcripts from CRPV-induced tumor tissues.** (A) The genomic positions of the oligo antisense probes (arrows) used in the Northern blot analyses and the antisense probes (brown lines) used for RNAscope RNA-ISH detection of viral transcripts (Fig 7), relative to the mapped CRPV promoters (P), polyadenylation signal (pA) and cleavage (CS) sites, RNA splice sites, and annotated viral ORFs. (B) Detection of CRPV transcripts from Hershey CRPV-induced wart tissue. Total RNA (5 μg) isolated from the tumor tissues (CRPV), or mock-infected normal skin tissues (Mock) was examined by Northern blot using the indicated ${}^{32}$P-labeled antisense oligoprobe specific to CRPV RNA above each blot. The arrows on the right side indicate the viral transcript with a size corresponding to detected bands (see Fig 5 and S5 Table for the details). Ethidium Bromide staining was used to detect 18S rRNA, which served as a loading control. The red color indicates the selected viral transcripts commonly detectable in CRPV tumor tissues (see Fig 5C for details).

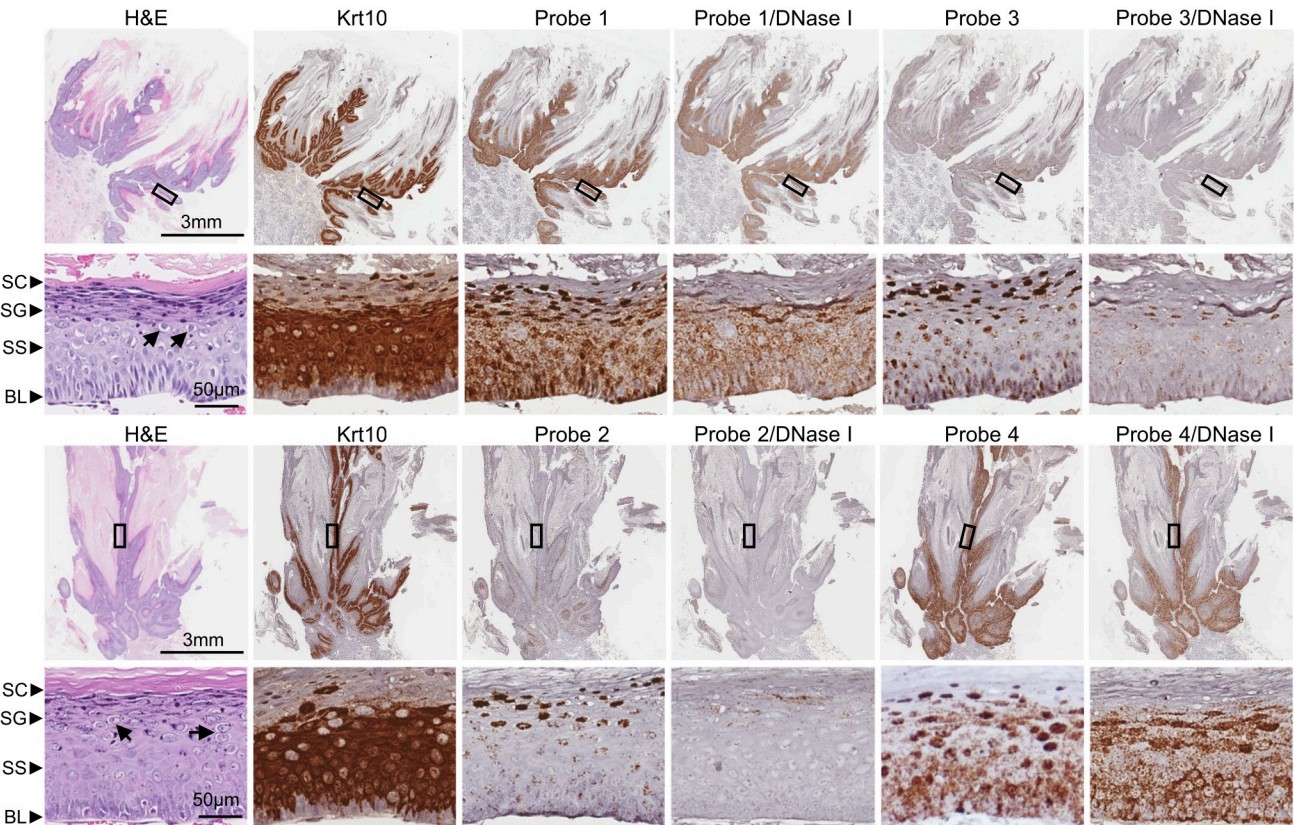

**Fig 7. The differentiation-dependent expression of CRPV transcripts in stratified epithelium within Hershey CRPV-induced tumor tissues determined by RNA _in situ_ hybridization (RNA-ISH).** The different layers of the epithelium, including the basal layer (BL), _stratum spinosum_ (SS), _stratum granulosum_ (SG), and _stratum corneum_ (SC), were identified using hematoxylin and eosin (H&E) staining, with arrows indicating the koilocytes. To detect individual viral transcripts, RNAscope RNA-ISH was performed on formalin-fixed paraffin-embedded (FFPE) tissue sections using the indicated probes antisense specific to the early (E6/E7 [Probe 1], E2/E1 [Probe 2], E4/E8/E2 [Probe 4]) or late L1/L2 [Probe 3]) region of the viral genome (See Fig 6A). The RNAscope RNA-ISH was performed in parallel with the adjacent tissue sections without or with prior DNase I treatment to remove the probes' cross-hybridization to a DNA strand. The probe targeting the host Krt10 transcript was used as a positive control.

The Hershey CRPV wart tissues showed an overt appearance of wart scars (S2A Fig), and the tissues in H&E staining displayed evidence for typical hyperplasia, koilocytosis with many koilocytes (arrows) and some fibrillary projections (Figs 7 and S2B–S2D). As shown in Fig 7, Krt10 RNA in this RNAscope RNA-ISH appeared predominantly in both lower and upper spinous but not basal layers of the tissues. As expected after DNase treatment of the tissue sections, most of the viral early transcripts from the early promoters $P_{90}$, $P_{156}$, and $P_{907}$ were detectable predominantly in the cytoplasmic compartments of keratinocytes by both Probe 1 and Probe 4 from basal, spinous, and granular layers of the wart tissues, but the Probe 4 also detected the viral late transcripts from the promoter $P_{1225}$ mainly for E1^E4 expression by using viral $pA_E$ for polyadenylation and the late transcripts from the promoter $P_{7525}$ splicing over to this region for L1 and L2 expression. Viral late transcripts from the promoter $P_{7525}$ and $P_{1225}$ and possibly from the putative promoters $P_{4258}$ and $P_{5765}$ for L2 and L1 expression can be detected mostly in the granular layers only by Probe 3 from the tissues with DNase treatment. A few of L1/L2 signals presented in the basal and spinous layers and detected by Probe 3 in the tissues after DNase treatment could be those double-spliced early transcripts or the early $pA_E$ readthrough transcripts (Fig 5C). As predicted, the Hershey CRPV wart tissues with

DNase treatment expressed only a few detectable E1 and E2 transcripts in the basal and spinous layers by Probe 2 targeting the E2 and E1 RNA regions as most parts of this region in the early transcripts are spliced out. A relatively stronger E1 and E2 region signals in the granular layers in this RNA-ISH detection was surprising, as reported [32], although these signals could be the remaining viral DNA left from incomplete DNase digestion. These RNA signals could also be those $P_{1225}$ late transcripts that underwent alternative RNA splicing to the nt 3065 splice acceptor site, which might have the potential for translation of E2 protein as reported in the upper layers of human cervical intraepithelial neoplasia (CIN) tissues [91].

For the Hershey CRPV wart tissue sections without DNase digestion, viral DNA in vegetative replication was detected mainly in the granular and cornified layers by all RNAscope RNA-ISH probes (Fig 7). We found that Probe 2 from the E2 region and Probe 3 from the L1 region were two probes for the best detection of viral DNA in the tissues without DNase treatment (Fig 7). Because most of the early transcripts in the E2 region are spliced and L1 transcripts are only minimal (Figs 1C and S3), the RNAscope probes in these two regions could detect only a few viral RNA transcripts from the tissues with DNase digestion, which was in sharp contrast to the viral DNA detection in the same tissue sections without DNase digestion (Fig 7). The tissues without DNase treatment showed some signals in the spinous or even basal layers (see all probes in Fig 7), most likely representing the viral episomal DNA in low copy number for initial CRPV infection and maintenance before reaching a vegetative replication level in the granular and cornified layers.

## CRPV protein sequence, structure, and ectopic expression

Our analysis of coding potentials of the identified CRPV transcripts revealed that the Hershey CRPV genome may encode 12 viral proteins (Fig 5A and S6 Table), including LE6, SE6, E10, E7, E1, E2, L2, L1 and other four proteins from the spliced ORF (E1^E4, E8^E2, E1^M1, and E1^M2) (Fig 5A). Among them, the structures of ten CRPV proteins have been predicted with AlphaFold2 (https://deepmind.google/technologies/alphafold/) (S8 Fig) and five of them can be modeled to the corresponding papillomaviral protein structures available in the literature, including CRPV LE6 to HPV16 E6, E7 to HPV1a E7, E1 to HPV18 E1, E2 to HPV16 and HPV11 E2, and L1 to HPV11 L1 (Fig 8). Data indicate the high domain conservations to other HPV proteins, but also some unusual intrinsically disordered or missing regions of unknown function in CRPV proteins (Figs 8 and S9).

CRPV has been thought to encode LE6 (273-aa) and SE6 (176-aa) (S6 Table). The LE6 differs from SE6 by 97-aa residues from its N-terminus and shares the same C-terminal sequences with the SE6, which is presumably translated from an in-frame AUG at nt 445, while an AUG at nt 154 is used for translation of the LE6. However, they may be encoded separately from two different viral early transcripts: the transcripts from a minor $P_{90}$ promoter for LE6 and the transcripts lacking the region covering the AUG at nt 154 from a major $P_{156}$ promoter for E10 and SE6 based on the mapped TSS in this report (Figs 2 and 5). Analysis of the Kozak content surrounding the AUG indicates that LE6 and E10 have a better Kozak consensus sequence than SE6.

We next cloned each of the predicted CRPV proteins (except E1^M1 and E1^M2) as a C-terminally FLAG-tagged protein for transient expression in rabbit RK13 and human HEK293T cells (Fig 9A) in aims to compare their expression capacity in these two types of cells under a heterologous strong cytomegalovirus (CMV) immediate early (IE) promoter to avoid differentiation-dependent expression. Twenty-four hours after transfection, the cells were harvested for Western blot or examined by immunofluorescent staining using an anti-FLAG antibody. As shown in Fig 9B, comparative expression of the LE6 and SE6 in both RK13

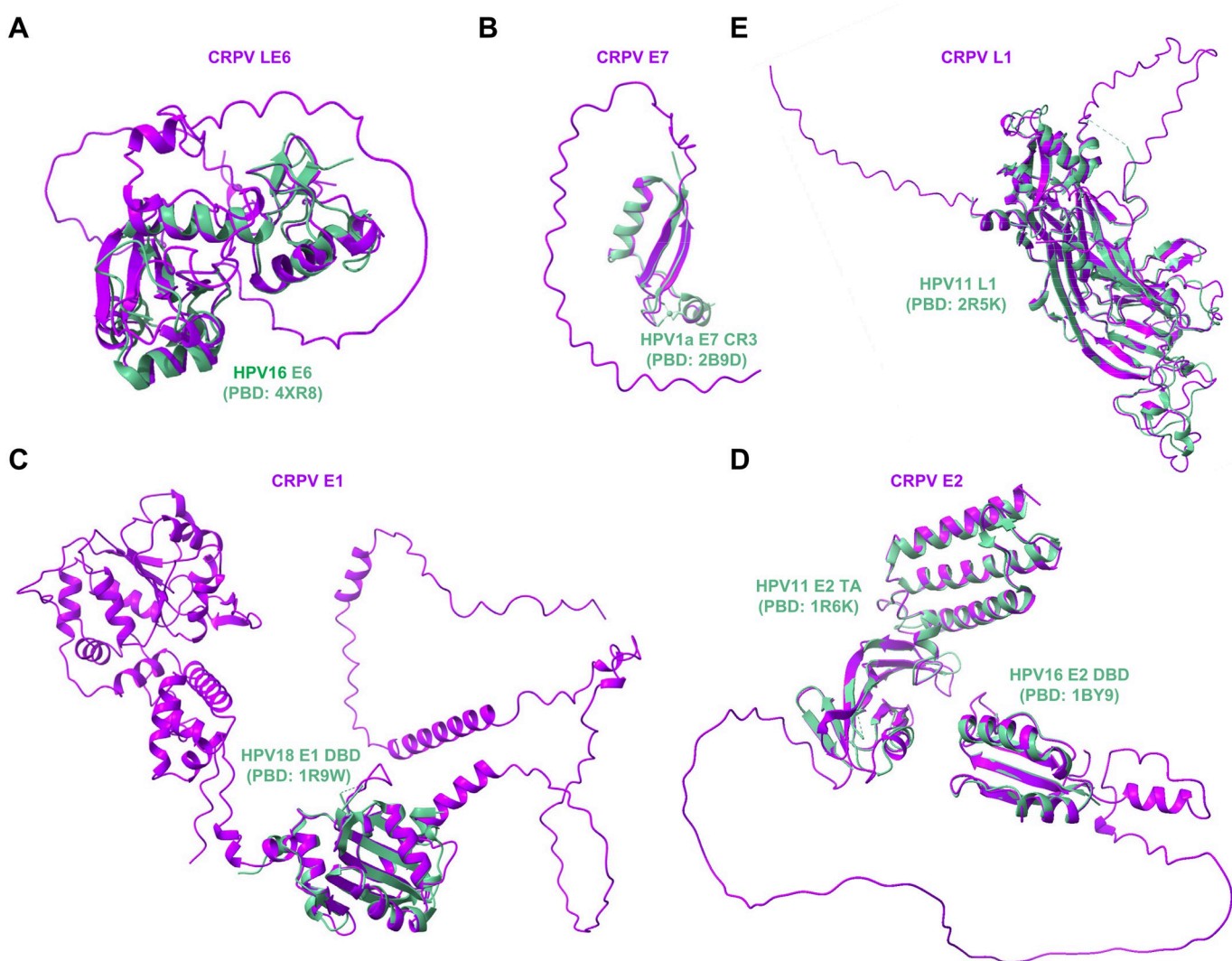

**Fig 8. CRPV proteins contain the conserved structural domains from HPV homologues.** The Alphafold 2 (https://deepmind.google/technologies/alphafold/) predicted structures of CRPV proteins (purple) were compared with the other experimentally determined structures of HPV proteins (green). The overlaps of the predicted CRPV protein structures with available HPV 3D structures generated by the UCSF ChimeraX v1.8 Matchmaker function include CRPV LE6 (A) with HPV16 E6 (PDB: 4XR8) [105], CRPV E7 (B) with HPV1a E7 CR3 domain (PDB: 2B9D) [106], CRPV E1 (C) with HPV18 E1 DNA-binding domain (DBD, PDB: 1R9W) [107], CRPV E2 (D) with HPV11 E2 transactivation domain (TA, PDB: 1R6K) [108] and HPV16 E2 DBD (PDB: 1BY9) [109], and CRPV L1 (E) with HPV11 L1 monomer (PDB: 2R5K) [110]. The shown models are not in scale.

and HEK293T cells was carried out in the presence or absence of proteasome inhibitor MG132 and revealed a higher level of LE6 expression in HEK293T cells than in RK13 cells by Western blot analyses, whereas the SE6 with a weak Kozak sequence surrounding the AUG codon was much less expressible in HEK293T cells and had no expression in RK13 cells. By calculation from two experiments, we found MG132 treatment led to a ~2.7-fold increase of LE6 level in RK13 and a ~1.8-fold increase of LE6 in HEK293T, while the SE6 expression in HEK293T cells could be detected only in the presence of MG132 (Fig 9B). An increased expression of LE6 and SE6 in the presence of MG132 suggests that both LE6 and SE6 are subject to proteasome-mediated degradation. Expansion of this comparative study for the expression of E1, E2, E7, E10, E1^E4, E8^E2, L1, and L2 did not show any major differences in the expression of

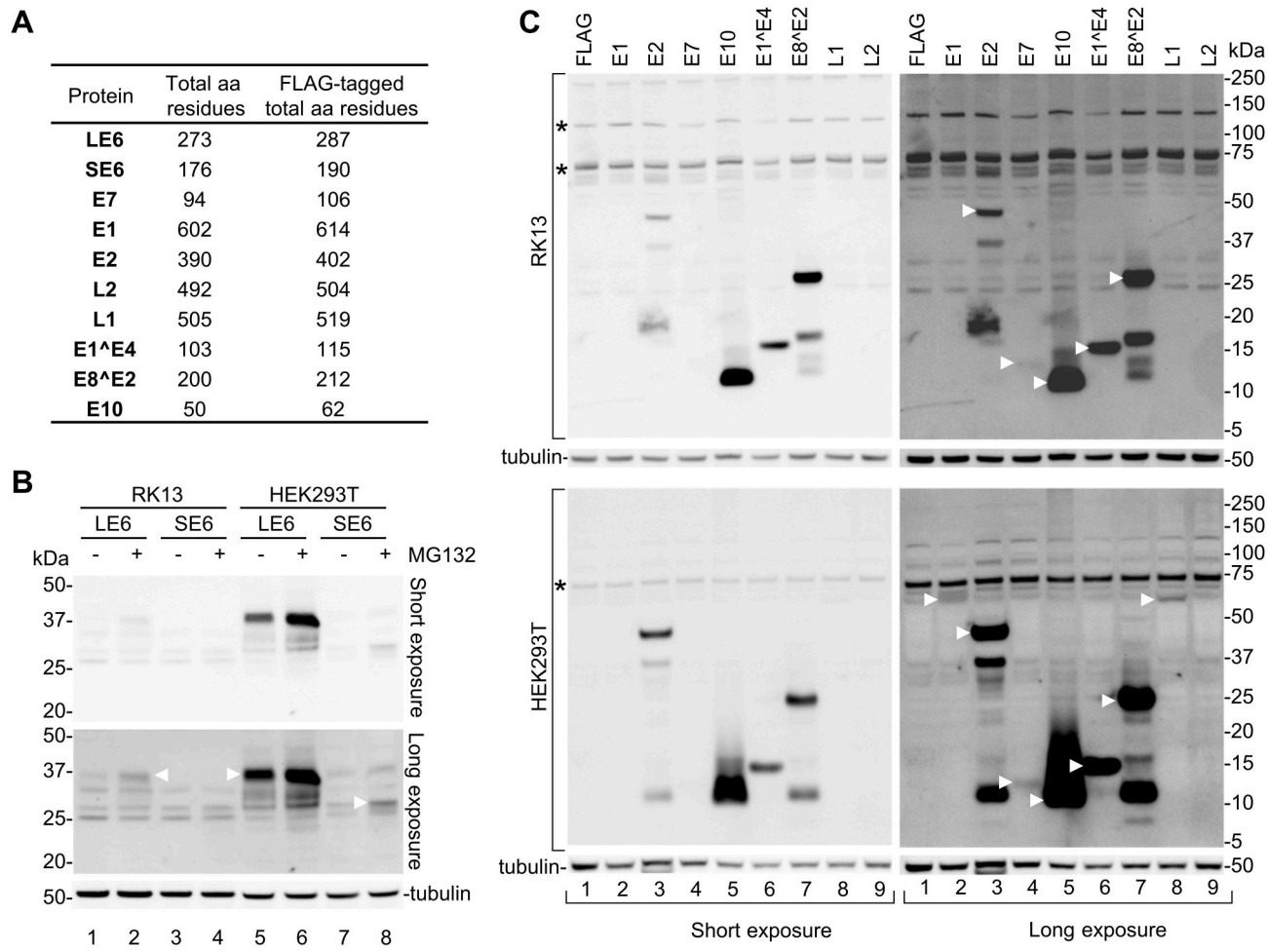

**Fig 9. Ectopic expression of FLAG-tagged CRPV proteins in RK13 and HEK293T cells.** (A) CRPV proteins with and without a FLAG-tag and their corresponding amino acid (aa) residues. (B) Comparative expression of CRPV long LE6 and short SE6 in RK13 and HEK293T cells in the presence or absence of proteasome inhibitor MG132. Total protein was extracted 24 h after cell transfection with the indicated CRPV protein expression vectors and examined by Western blot using an anti-FLAG antibody. (C) Comparative expression assays of CRPV E1, E2, E7, E10, E1^E4, E8^E2, L1, and L2 in RK13 and HEK293T cells. Total protein extracts at 24 h after cell transfection were immunoblotted by an anti-FLAG antibody. White triangles point to the protein bands of the expected size. The asterisks mark non-specific protein bands. Tubulin in panels B and C was blotted for protein loading controls. One blot is shown as a representative of three experiments.

CRPV proteins from RK13 to HEK293T cells (Fig 9C), except for E1 and L1 of which minimal expression was detected only in HEK293T cells (Fig 9C, lanes 2 and 8). L2 was not expressible in either of the tested cell lines. In the case of E2 and E8^E2, two additional smaller protein bands were detected (Fig 9C, lanes 3 and 7), most likely from alternative RNA splicing when these regions were inserted into an expression vector as observed for other viral ORF [92].

Analysis of subcellular localization of FLAG-tagged CRPV proteins in both RK13 and HEK293T cells was performed by immunofluorescent staining using an anti-FLAG antibody (Figs 10 and S10). In both cell lines, LE6 and SE6 showed a nuclear expression (Figs 10 and S10). The LE6 was found predominantly in the nucleus in COS-7 cells, but SE6 as a soluble, serine-phosphorylated cytoplasmic protein [35]. In HEK293T cells, we were able to detect predominant nuclear expression of E1, E2, E7, E8^E2, and L1, whereas E7 and E1^E4 showed both nuclear and cytoplasmic expression and E10 mainly cytoplasmic expression (Fig 10).

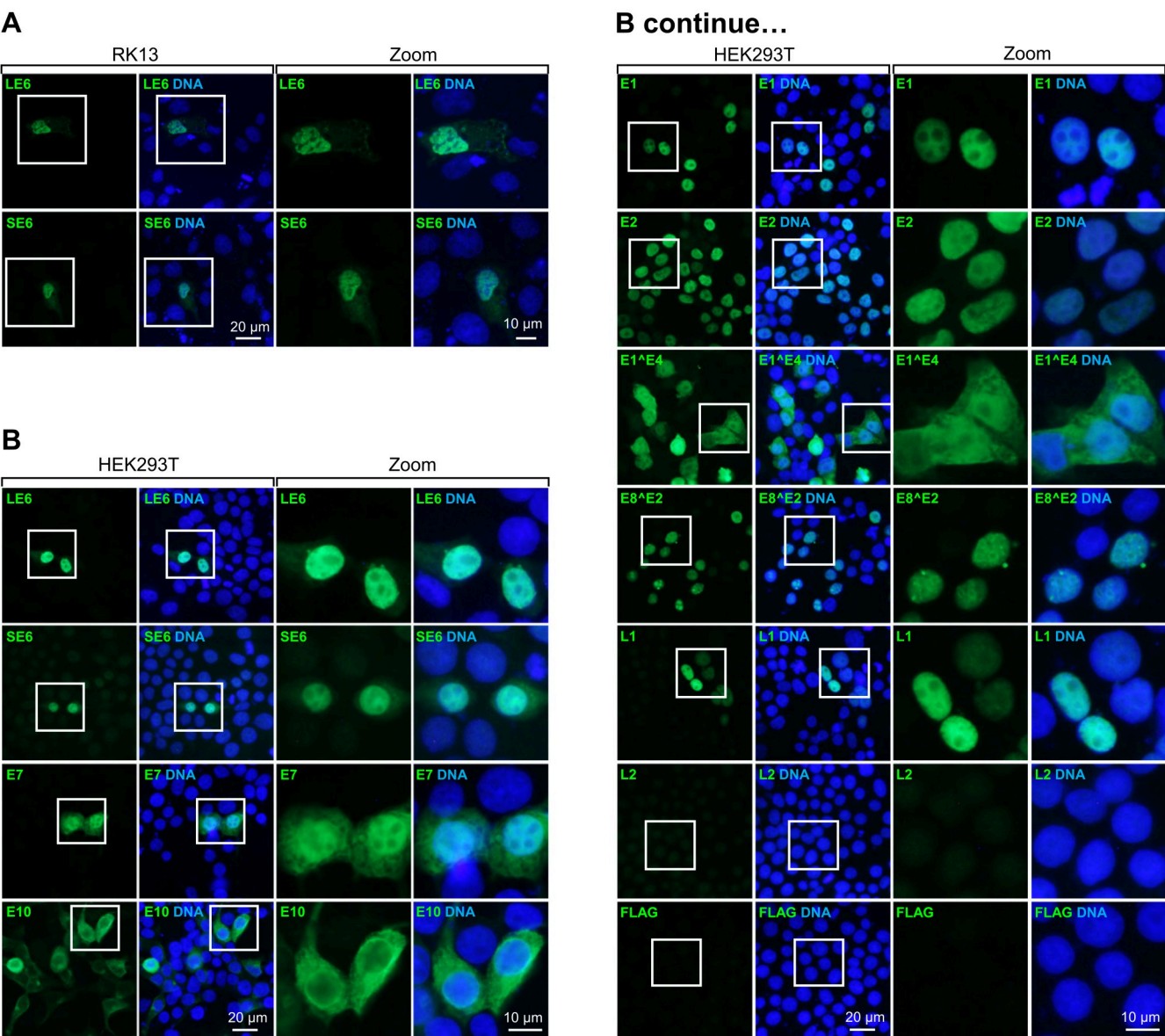

**Fig 10. Ectopic expression and subcellular localization of CRPV proteins determined by immunofluorescent staining.** Immunofluorescent staining of FLAG-tagged CRPV proteins ectopically expressed in RK13 (A, S10 Fig) and HEK293T cells (B) was examined using an anti-FLAG antibody 24 h after cell transfection with the indicated CRPV protein expression vectors. The cell nuclei (blue) were counterstained using Hoechst 33342 dye.

Consistent with Western blot, we could not detect L2 protein expression in HEK293T cells (Fig 10). Similar localizations of expressible CRPV proteins (E2, E7, E10, E1^E4, E8^E2) were observed in RK13 cells, but E1, L1, and L2 were not expressible in RK13 cells (S10 Fig).

## Discussion

Since the discovery of a filterable agent capable of inducing papillomas in cottontail rabbits in 1933 [1,2], studies on CRPV, the first mammalian DNA tumor virus and a founding member in the papillomavirus family [93,94], have made many important contributions to our understanding of virus-host interactions, viral immunology, tumor pathology, and tumorigenesis

[6,12]. A whole CRPV genome sequence has been decoded for Shope CRPV (GenBank NC_001541.1) [46], CRPV subtype a and b (GenBank AJ404003 and AJ243287) [95], and CRPV Hershey strain (GenBank JF303889.1) [47]. However, the lack of a carefully defined CRPV transcription map has been a challenge for us to understand the molecular mechanisms of CRPV infections and pathogenesis in the past 90 years, in addition to lacking CRPV cell culture systems and rabbit reagents. In this report, we have constructed for the first time a full transcription map of the CRPV genome from Hershey CRPV-induced papillomas. We identified that the Hershey CRPV expresses its early viral transcripts from three promoters, $P_{90}$, $P_{156}$, and $P_{907,}$ and late transcripts mainly from two promoters, $P_{7525}$ and $P_{1225}$. By using alternative RNA splicing of seven splice sites (four 5′ splice sites or donor sites and three 3′ splice sites or acceptor sites), viral early and late transcripts undergo splicing regulation and polyadenylation either at a viral early pA site or late pA site for CRPV gene expression and encode 12 putative viral proteins from approximately 33 viral RNA isoforms during CRPV infection.

CRPV E6 ORF encodes LE6 of 273-aa residues and SE6 of 176-aa residues separately from its first and second AUGs [33,35]. ORF E10 is colinear with the ORF E6 in a different frame, which encodes 50-aa residues. As both LE6 and SE6 are undetectable during CRPV infection, full-length E6 ORF in an expression vector was found to express primarily LE6 but little SE6 [33]. Both CRPV LE6 and SE6 in NIH 3T3 cells have transformation activities, but E10, formerly E8, does not [48]. In our constructed CRPV transcription map, the promoter $P_{156}$ activity was five-fold higher than promoter $P_{90}$ activity in Hershey CRPV-induced papillomas (Fig 2C and 2D). Thus, the $P_{156}$-derived transcripts, which are abundant in the wart tissues or virus early infection, would encode predominantly E10 (nt 323–475) or SE6 (nt 455–975) but not the LE6 (nt 154–975) because these $P_{156}$-derived transcripts are missing the first AUG at the nt 154 position. This prediction is consistent with the assumption that the major E6 protein in CRPV-induced tumors is most likely the SE6 protein initiated at the second AUG [33]. However, due to the weak Kozak content of the second AUG, the SE6 expressible levels may be low or minimal, as we found (Fig 9B). Interestingly, the promoter $P_{90}$-derived early transcripts D and G (Fig 5) were detectable by Northern blot (Fig 6, lane 1) and a CRPVLE6ATG-knockout virus was found to be dysfunctional in growing tumors [44], suggesting the presence of a functional LE6. In support of this assumption, a CRPVSE6ATG-knockout virus was functional in growing tumors, but the tumor size was significantly smaller [44].

Hershey CRPV genome has a TTT codon insertion after nt 4342, but an ACA codon deletion after nt 4362, and a CA dinucleotide insertion after nt 4371 all in the predicted E5 ORF region when compared with the Shope CRPV genome (S1 Fig). The CA Insertion after nt 4371 in Hershey CRPV creates a frameshift of the E5 ORF to encode a truncated E5 of 62-aa residues that contrasts to the 101-aa residue E5 of Shope CRPV (S11 Fig). Transformation activities of full-length CRPV E5 appear debatable. In NIH 3T3 cells, the Shope CRPV E5 has no transformation activity [39], but in C127 cells, CRPV E5 does so, with weak tumorigenicity in nude mice [63]. Others also showed that the putative E5 ORF in CRPV is dispensable for papilloma formation in domestic rabbits [42, 44]. The location of CRPV E5 ORF unusually overlaps the L2 N-terminal ORF and contains two AAUAAA PAS, respectively, at nt 4351 and nt 4470. In the constructed transcription map of Hershey CRPV in this report, the mapped viral early polyadenylation site at nt 4368 by using the PAS at nt 4351 is located within the E5 ORF (Fig 3), of which usage for viral early RNA polyadenylation will disrupt the E5 ORF. We did not find any usage of the PAS at nt 4470, although the transcription of a truncated E5 mRNA from Hershey CRPV infection might hypothetically use the PAS at nt 4470 for polyadenylation. Others may argue that the E5 ORF might present on L2 transcripts, but L2 translates from a different frame starting within the putative E5 ORF and bears a strong Kozak sequence (. . .. RccAUGG. . ..), with a purine both in -3 and +4 AUG positions [96]. Unfortunately, the

putative E5 ORF does not have a strong Kozak sequence, but a . . ..ttttttAUGG. . ... sequence. This feature of E5 on L2 transcripts will not make E5 translatable, but rather prefer L2 translation. Given that CRPV E5 is not absolutely required for papilloma formation in domestic rabbits, we agree with the early suggestion that the reading frame recognized as the putative E5 ORF of CRPV may not be the bona fide E5 gene [42]. Thus, the E5 ORF is deleted from the transcription map constructed from Hershey CRPV-induced tumor tissues.

Unlike cottontail rabbits, that are natural host for CRPV infection and produce high titers of CRPV virions, NZW domestic rabbits are not the natural host for CRPV infections and only produce very low levels of viral particles despite detectable by anti-L1 antibodies generated in CRPV-infected animals [97]. CRPV late transcripts L2 and L1 are minimal in papilloma tissues and approximately ~40-fold less than the detectable viral early transcripts. These findings further suggest that low levels of L1 and L2 transcripts may lead to low viral production. These late transcripts are mainly derived from two late promoters $P_{7525}$ and $P_{1225}$, but some appear from the promoter $P_{907}$, which is more responsible for the viral early transcripts to encode E7 (Fig 2C and 2D). The $P_{907}$ remaining in activation in the late stage of CRPV infection may lead to the distribution of the detectable E7 RNA from the basal to granular layers (Fig 7) and a high level of E7 mRNA in the upper differentiated epithelial layers [32], which could be important for vegetative viral DNA replication. The $P_{1225}$ TSS heterogeneity is attributable to the lack of a consensus TATA box in the viral late promoter in the E7 ORF region. Instead, the presence of a TATA-like element (TATT or TTTA, etc.) is characteristic of other known viral late promoters in HPV18 [76], MmuPV1 [80], and KSHV K8.1 [98]. Interestingly, two other weak promoter activities appear detectable with one at nt 4258, upstream of L2 ORF, and the other at nt 5765, within the L2 ORF, by extremely high-throughput PacBio Isoseq (S5A Fig). The weak late promoter activities in the corresponding viral genomic regions had been proposed before in the E4 and E5 ORF regions for HPV16 in W12 cells [99] and in the E4 ORF region for HPV31 in CIN612 9E-derived raft cultures [78].

Our successful construction of this complete CRPV transcription map from the Hershey CRPV-infected lesion in this study is significant as it will lay a solid foundation for many future studies using this sophisticated model. The CRPV/rabbit model induces infection with long-term persistence and malignant progression of lesions with consistency and reproducibility [8,100]. It is one of the surrogate models for developing and testing novel anti-tumor and anti-viral compounds against high-risk HPV infections [7,101,102]. More importantly, we and others have generated a large panel of functional mutant CRPV genomes over the years [6,12]. Together with CRPV transcription map generated in the current study, we will be able to answer many key questions, including the changes in cell differentiation-dependent expression of viral genes in correlation to CRPV lesion progression and pathogenesis and timing for novel treatments of HPV-associated diseases and cancers.

## Materials and methods

### Ethics statement

All animal experiments were performed using the animal protocol approved by the Institutional Animal Care and Use Committee of The Pennsylvania State University College of Medicine (PRAMS200747110, approved 05/06/2019) according to NIH guidelines for the care and use of animals in this research.

### CRPV infection

Four male and four female New Zealand White (NZW) rabbits were experimentally infected with CRPV Hershey strain, as described previously [11,43]. Briefly, the rabbit's back skin was

shaved and pre-wounded with a scalpel blade. Three days later, serial dilution of CRPV inocula was applied to the back sites, as shown in Fig 1A. Two sites treated with the same inoculum volume of sterile 1× PBS were used as negative controls. The induced wart tissues were measured weekly and collected eight weeks after the infection for RNA-seq analysis.

## Total RNA-seq and bioinformatics analysis

The collected wart tissues were homogenized in TriPure reagent (Roche), and total RNA was extracted according to the manufacturer. To eliminate viral DNA, the obtained RNA was on-column DNase-treated using an RNeasy Kit (Qiagen). The RNA concentration and integrity were assessed by Bioanalyzer 2100 (Agilent). The total RNA sequence libraries were prepared using Illumina Stranded Total RNA (Illumina, RS-122-2201) protocol with TruSeq V4 chemistry and sequenced with 2×125 modality and depth of 100 million reads per sample. The obtained reads were trimmed of adapters and low-quality bases and aligned to chimeric rabbit (*Oryctolagus cuniculus*)/CRPV Hershey strain reference genome (GenBank Acc. No. JF303889; GI: 333906187) permutated at nt 7421 using STAR aligner package [103]. These RNA-seq data had been deposited in NCBI's Gene Expression Omnibus with an accessible number GSE124211. The Integrative Genomics Viewer (IGV, Broad Institute) program was used to visualize reads coverage. The viral SJs were identified by STAR aligner with additional criteria: (1) a presence of the canonical splice sites; (2) the number of uniquely mapped reads is greater than the number of multimapped reads. A Sashimi plot for SJ visualization was generated by IGV.

## Rapid amplification of cDNA ends (RACE) and PacBio Iso-seq sequencing

The 5′ and 3′ RACE assays were carried out using SMARTer RACE 5′/3′ Kit (Takara) according to the manufacturer's instructions using 1 μg/reaction of total RNA as the template. The primers used in the assays are in the S7 Table. The final PCR products were gel purified, cloned into pCR2.1-TOPO vector (Thermo Fisher Scientific), and sequenced by Sanger sequencing. To obtain the comprehensive coverage of viral TSS, the 5′ RACE products were subjected to single molecule, real-time (SMRT) sequencing using PacBio Iso-seq-technology. The sequencing libraries were prepared using SMRTbell Template Prep Kit 1.0 (Pacific Biosciences) and sequenced on PacBio RS II sequencer. The obtained sequence reads were trimmed of adaptors and mapped to the linearized CRPV Hershey strain reference genome. The position of read's 5′ ends were extracted and quantified as TSS for further analysis.

## RT-PCR

To remove contaminating genomic DNA, total RNA was treated with a TURBO DNA-free Kit (Ambion). Reverse transcription (RT) was performed with the SuperScript II kit (Thermo Fisher Scientific). Amplification of reversed transcribed cDNA was performed by PCR using the AmpliTaq Polymerase Kit (Thermo Fisher Scientific) according to the manufacturer's protocols. The CRPV Hershey strain-specific primers used to detect viral transcripts are described in the S7 Table. PCR amplifications were performed under the same conditions: a primary denaturation step at 94°C for 2 min, followed by 35 cycles of 1 min at 94°C, 2 min at 55°C and 2 min at 72°C, and final extension for 7 min at 72°C.

## Northern blot analysis

Total RNA (5 μg) isolated from CRPV-induced skin tumors was mixed with formaldehyde load dye (Thermo Fisher Scientific) and denatured at 75°C for 15 min. The RNA samples were

then separated in 1% (wt/vol) formaldehyde-containing agarose gels in 1× morpholine propane sulfonic acid (MOPS) running buffer and transferred onto GeneScreen Plus hybridization transfer membrane (Perkin Elmer). After UV light crosslinking, the membrane was prehybridized with PerfectHyb Plus hybridization buffer (Sigma-Aldrich) for 2 h at 42°C followed by overnight hybridization with $^{32}$P-labeled CRPV-specific oligo probes (S7 Table). After hybridization, the membrane was washed once with a 2 × SSPE/0.1% SDS solution for 5 min at room temperature and twice with 0.2 × SSPE/0.1% SDS for 15 min at 42°C and then exposed to phophorscreen and X-ray film for signal detection.

### RNA *in situ* hybridization (RNA-ISH)

For *in situ* detection of viral transcripts, the tissues were fixed in 10% neutral buffered formalin for 20 h at room temperature, dehydrated, and embedded in paraffin. The sections were cut into the series of 5 μm slides and subjected to RNA-ISH using RNA-scope technology (Advanced Cell Diagnostics, ACD) as recommended. Four custom-designed RNAscope (ACD) probes derived from CRPV Hershey strain genome were used: Probe 1 (211–959); Probe 2 (3113–3691), Probe 3 (5831–6847), and Probe 4 (3715–4010). The probe targeting rabbit keratin 10 (RNAscope Probe-Oc-KRT10, Cat No. 549841, ACD) was used as a positive control. The signal was detected by colorimetric staining using RNAscope 2.5 HD Assay-BROWN (ACD) followed by hematoxylin Gill's No. 1 solution (Sigma-Aldrich) counterstain. The slides were dehydrated, mounted in Cytoseal XYL (Thermo Scientific), and scanned at 40× resolution using Aperio CS2 Digital Pathology Scanner (Leica). DNase digestion was performed before RNA-ISH, as previously described in [80].

### CRPV ORF cloning and ectopic expression

To express viral proteins, the cDNAs of individual ORF with optimized Kozak sequence were amplified by RT-PCR from total RNA isolated from infected tissues and cloned into pFLAG-CMV-5.1 (Sigma-Aldrich). The obtained plasmid DNAs (2 μg) were transfected into rabbit RK13 (CCL-37, ATCC) or human HEK293T cells (CRL-3216, ATCC) plated in a 6-well plate in Dulbecco's Modified Eagle Medium (DMEM, Thermo Fisher Scientific) supplemented with 10% fetal bovine serum (FBS, Cytiva). All transfections were carried out using LipoD293 transfection reagent (SignaGen Laboratories). The expression of viral proteins was monitored by Western blot and immunofluorescence staining 24 h after transfection. Alternatively, the cells were treated with 15 μM of proteasome inhibitor MG132 (M7449, Sigma) for 4 h before harvesting. The control cells were treated with vehicle (DMSO) only for the same time period.

### Western blot

The total protein extracts were prepared 24 h after transfection by direct lysis of transfected cells in 2× SDS protein sample buffer with the addition of 5% 2-mercaptoethanol. After SDS-PAGE, the expression of individual viral proteins was determined by Western blot with a rabbit anti-FLAG antibody (F7425, Sigma) in combination with the peroxidase-conjugated secondary antibody (A0545, Sigma). The signal was developed by SuperSignal West Pico PLUS Chemiluminescent Substrate (Thermo Fisher Scientific) and captured by ChemiDoc Touch (Biorad). The membranes were then stripped and reprobed with mouse anti-β-tubulin antibody (T5201, Sigma) for loading control.

## Immunofluorescent staining

Before staining, the plasmid-transfected cells growing on the coverslips were fixed with 4% paraformaldehyde (Electron Microscopy Sciences) for 20 min at room temperature, quenched with 100 mM glycine (Sigma-Aldrich) for 5 min, and permeabilized with 0.5% Triton X-100 (Promega) for 15 min. The cells were then blocked with 3% Blot-Qualified BSA (W3841, Promega) in PBS supplemented with 0.05% Tween-20 (TPBS) and stained with rabbit anti-FLAG antibody (F7425, Sigma) in combination with AlexaFlour488-labeled anti-rabbit secondary antibody (Thermo Fisher Scientific). The cell nuclei were counterstained by Hoechst 33342 dye (Thermo Fisher Scientific).

## Protein sequence alignment and structure prediction

The protein sequences were aligned using ClustalOmega (https://www.ebi.ac.uk/Tools/msa/clustalo/). The collab AlphaFold 2 [104] was used to determine the 3D structure of all identified viral proteins (S1 Data). The highest-scoring models were visualized by UCSF ChimeraX v1.8 (https://www.cgl.ucsf.edu/chimerax/). The ChimeraX' Matchmaker function was used to compare the predicted CRPV proteins and the corresponding HPV proteins with published experimental structures available in the RCSB Protein Data Bank (https://www.rcsb.org/).

## Supporting information

**S1 Fig. Observed sequence differences from the Hershey CRPV genome (JF303889.1) to the Shope CRPV genome (NC_001541).** The nucleotide position, nucleotide change, locus, and type of mutations were detected by sequence alignment of the Hershey and Shope reference CRPV strains.
(TIF)

**S2 Fig. Histology of CRPV-induced wart tissues.** (A) The morphological appearance of CRPV-induced wart tissues on the back of four female (F) rabbits used in the study 8 weeks post inoculation with CRPV Hershey strain. R1-5 and L1-5 represent five sites inoculated with a 10-fold dilution of the virus inoculum on the right (R) or the left (L) side of the rabbit back. The wart tissues used for RNA extraction and RNA-seq are marked in red. (B-E) Histology of Hershey CRPV-induced wart tissues. On low magnification (B, 20×), these verrucous lesions form papillary projections of hyperplastic squamous epithelium covered by thick bands of lamellated keratin and variably supported by narrow fibrovascular cores. On higher magnifications (C, 40×), there is marked hyperkeratosis in addition to a severe expansion of the stratum basales with increased mitotic activity as well as thickening of the *stratum spinosum* (acanthosis) and *stratum granulosum*. Squamous epithelial cells in the superficial layers frequently display abnormally large nuclei surrounded by large, clear perinuclear halos (koilocytosis/koilocytotic atypia D, 60×, arrows). The superficial dermis is infiltrated by large numbers of lymphocytes in some samples (E, 40×, arrows). Histopathological features of malignancy were not identified.
(TIF)

**S3 Fig. Distribution of RNA-seq reads across CRPV genome.** (A) The depth of RNA-seq reads coverage across the Hershey CRPV genome (GenBank Acc. No. JF303889.1) linearized at nt 7421 in one representative tissue sample (R4941) with a decreasing reads scale shown in the upper right corner. (B) The distribution of RNA-seq reads mapped to both (shown in blue), sense (shown in red), and antisense strand (shown in pink) of the CRPV Hershey genome visualized by IGV using the autoscale shown in the upper right corner.
(TIF)

**S4 Fig. Consistency of detected RNA transcription start sites, polyadenylation cleavage sites, and newly identified splice sites from CRPV tumor tissues from four separate animals.** (A) The 5′ RACE for mapping transcription start sites. (B) The 3′ RACE for mapping RNA polyadenylation cleavage sites. (C) Validation by RT-PCR of newly identified splice sites for CRPV L1 expression. All assays were performed on cDNA generated from total RNA from four different animal tumor tissues used for RNA-seq. The primer sequences are in S7 Table. (TIF)

**S5 Fig. Mapping and usage of CRPV promoters.** (A) The distribution of RNA-seq reads from one representative sample (R4941, upper panel) and viral transcriptional starting sites (TSS) determined by PacBio Iso-seq of 5′ RACE products obtained by primer antisense to various parts of the viral genome (black arrows beneath the diagram). The blue arrows at the top indicate the position of the major viral promoters (see Fig 2C). The black boxes below represent annotated viral ORFs. (B) The percentage of PacBio long reads from individual 5′ RACE libraries mapped to individual viral promoters (see S3 Table for more details). (TIF)

**S6 Fig. The TATA or TATA-like sequence upstream of mapped CRPV promoters.** The nucleotide sequences of mapped CRPV promoter regions with predicted TATA or TATA-like box motifs positioned (indicated by arrows) upstream of the mapped transcriptional start site (TSS). (TIF)

**S7 Fig. The minor splicing junctions detected in CRPV transcripts.** (A) RNA-seq coverage of the representative samples with a Sashimi plot of detected splice junctions (SJ) shown as the numbered arches (numbers 1–12 in the filled boxes). The associated number represents the number of total splice junction reads detected in all four samples (see S4 Table). The highly abundant SJ (>50 reads) are shown in blue. The additional less abundant SJ (>10<50 reads) are shown in red. The positions of minor splice sites labeled below the Sashimi plot are shown in red. The red arrows below the assigned viral ORFs (black boxes) represent the primers used in RT-PCR. (B) The gel electrograph of the RT-PCR product obtained using the primers shown in (A). The individual products were gel-purified and subjected to Sanger sequencing. The chromatographs with detected SJ are shown below. Unspl., unspliced product. (TIF)

**S8 Fig. Predicted structures of CRPV proteins.** CRPV ORFs and their corresponding amino acid residues are shown in the S6 Table. The structure of each CRPV protein was predicted using the AlphaFold 2 algorithm with one representative model shown in the rainbow coloring with the N-terminus (N) marked as dark blue and the C-terminus (C) as dark red. The models shown are not in scale. See more details of the protein data bank (pdb) files in S1 Data for the individual protein structures predicted by AlphaFold 2. (TIF)

**S9 Fig. Alignment between predicted structures of CRPV proteins LE6 (A), E7 (B), E1 (C), E2 (D) and L1 (E) with the known structures of corresponding HPV proteins (see Fig 8).** The matched residues are shown in yellow boxes. Ca RMSD in grey columns represents the single-point spatial variation among residues. All sequence alignments were performed by UCSF ChimeraX v1.8 using the Matchmaker showAlignment function. (TIF)

**S10 Fig. Ectopic expression and subcellular localization of CRPV viral proteins in RK13 cells.** Rabbit kidney RK13 cells were transfected with the plasmids expressing FLAG-tagged

CRPV proteins. Twenty-four hours after transfection, the cells were fixed and stained with an anti-FLAG antibody (green). The cell nuclei shown in blue were stained with Hoechst 33342 dye.
(TIF)

**S11 Fig. Alignment of amino acid sequences of E5 proteins encoded by reference Shope (GenBank Acc. No. NC_001541) strain and probably by CRPV Hershey strain.** All sequence alignments were performed using the Clustal Omega software (www.ebi.ac.uk/Tools/msa/clustalo/).
(TIF)

**S1 Table. RNA-seq mapping summary from individual rabbits.**
(XLSX)

**S2 Table. Heterogeneity of the mapped transcription sites by 5′ RACE based on individual clone sequencing.**
(XLSX)

**S3 Table. Analysis of position and frequency of viral TSS based on Pac-Bio sequencing.**
(XLSX)

**S4 Table. Splice junctions (SJ) detected in CRPV RNA-seq by STAR aligner.**
(XLSX)

**S5 Table. The nucleotide position, size, and coding potential of predicted CRPV transcripts.**
(XLSX)

**S6 Table. The coding regions and amino acid sequences of the predicted CRPV proteins.**
(XLSX)

**S7 Table. The oligos used in the study.**
(XLSX)

**S1 Data. The Protein Data Bank (PDB) files of CRPV protein structures predicted by AlphaFold 2.**
(ZIP)

## Acknowledgments

We would like to thank the teams at the Center for Cancer Research Sequencing Facility (CCR-SF) for the library preparation and sequencing.

## Author Contributions

**Conceptualization:** Vladimir Majerciak, Jiafen Hu, Neil D. Christensen, Zhi-Ming Zheng.

**Data curation:** Pengfei Jiang, Vladimir Majerciak, Jiafen Hu, Zhi-Ming Zheng.

**Formal analysis:** Pengfei Jiang, Vladimir Majerciak, Thomas J. Meyer, Maggie Cam, Matthew Lanza, Zhi-Ming Zheng.

**Funding acquisition:** Jiafen Hu, Neil D. Christensen, Zhi-Ming Zheng.

**Investigation:** Pengfei Jiang, Vladimir Majerciak, Jiafen Hu, Karla Balogh, Debra Shearer, Matthew Lanza, Zhi-Ming Zheng.

**Methodology:** Pengfei Jiang, Vladimir Majerciak, Jiafen Hu, Karla Balogh, Debra Shearer.

**Project administration:** Jiafen Hu, Zhi-Ming Zheng.

**Resources:** Jiafen Hu, Neil D. Christensen, Zhi-Ming Zheng.

**Software:** Thomas J. Meyer, Maggie Cam.

**Supervision:** Jiafen Hu, Zhi-Ming Zheng.

**Validation:** Pengfei Jiang, Vladimir Majerciak, Jiafen Hu, Debra Shearer, Matthew Lanza.

**Visualization:** Pengfei Jiang, Vladimir Majerciak, Debra Shearer, Matthew Lanza.

**Writing – original draft:** Pengfei Jiang, Vladimir Majerciak, Jiafen Hu, Zhi-Ming Zheng.

**Writing – review & editing:** Vladimir Majerciak, Jiafen Hu, Neil D. Christensen, Zhi-Ming Zheng.

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
