## [Decision Letter · Decision Letter 0]

1 Jul 2024

Dear Dr. Zheng,

Thank you very much for submitting your manuscript "Full transcription map of Cottontail rabbit papillomavirus in tumor tissues" for consideration at PLOS Pathogens. As with all papers reviewed by the journal, your manuscript was reviewed by members of the editorial board and by several independent reviewers. In light of the reviews (below this email), we would like to invite the resubmission of a significantly-revised version that takes into account the reviewers' comments.

We cannot make any decision about publication until we have seen the revised manuscript and your response to the reviewers' comments. Your revised manuscript is also likely to be sent to reviewers for further evaluation.

Sincerely,

Paul F. Lambert

Academic Editor

PLOS Pathogens

Patrick Hearing

Section Editor

PLOS Pathogens

Michael Malim

Editor-in-Chief

PLOS Pathogens

orcid.org/0000-0002-7699-2064

Reviewer's Responses to Questions

**Part I - Summary**

Reviewer #1: Jiang et al. describe the complete transcript map of Cottontail rabbit papillomavirus using RNA derived from tumors induced in New Zealand White rabbits. They characterize transcription start sites, splice sites, and polyadenylation sites using a variety of standard mapping and state-of-the-art next generation sequencing methods. This results in an improved transcription map compared to the one provided on the PAVE web site. While this is impressive, the data mainly confirm findings (start sites, splice sites, differences of transcript levels) obtained in the past 35 years. The new findings are that CRPV also has a promoter in E7 driving late gene expression and its transcripts use a splice donor site in E4 and an acceptor site in front of L1. This study mainly confirms that human and animal PV have very predictable transcript patterns. The study lacks mechanistic insights and does not significantly enhance our understanding of PV. In summary, this a very well-executed study but it lacks novelty.

Reviewer #2: Analysing the transcriptome of cotton tail rabbit papillomavirus (CRPV) is important as it promises to add to our understanding of animal papillomaviruses. It will also allow comparison with human papillomaviruses, especially the cancer-causing ones. CRPV is an important model since it could be used as a in vivo model for testing therapies against human papillomaviruses and cancer progression. The manuscript contains for the first time a thorough analysis of the CRPV transcriptome. This information can be used in future to design new studies examining e.g. CRPV tumorigenesis. The results section follows the pattern of a previous transcriptomic study of the murine papillomavirus from this group. However, I believe the data could be curated better into far fewer figures, while the second part of the results in Figures 7, 9-11 is confusing in places and does not really add to the impact of this study.

Reviewer #3: In this study, the authors provide a comprehensive mapping study of the mRNA transcripts produced by the Cottontail Rabbit Papillomavirus. This now allows the position of the key viral promotors and polyA sites to be ascertained, and with the confirmation of splice donor and splice acceptor positions, it also allows the realisation of primary amino acid sequences. The authors have added tertiary structure predictions, and have also gone on to localise the expression patterns of some of the major mRNA species in CRPV-induced lesions using an RNA scope approach. In doing this, they have been careful to exclude signals derived from the possible amplification of viral episomes, which because of their abundance, can cause problems when trying to detect mRNA transcripts specifically in the upper epithelial layers. The expression studies in tissue culture provide a small amount of additional knowledge, with the main value being to provide data in support for previous work on CRPV protein function and tissue biology, and to pave the way for mor comprehensive studies on PV biology nd immunology in the future. This is a long overdue study, and the knowledge provided allowing a more substantial use of the CRPV system for basic science research and drug discovery in the future. The CRPV model is the best model of persistent infection in an immune competent host, and offers unique analysis opportunities when compared to the current mouse PV systems, which require immune deficient animals, and the rabbit oral papillomavirus system, which doesn’t support persistent infection. The Zheng group are world leaders in the analysis of PV RNA processing, and this study is of high quality.

**Part II – Major Issues: Key Experiments Required for Acceptance**

Reviewer #1: (No Response)

Reviewer #2: While it is clear that RNASeq was performed on tumors from 4 x male and 4 x female rabbits, it is not clear from the materials and methods section, or the figure legends, if the subsequent 5’-, 3’-RACE and RT-PCR data were carried out on more than one tumor tissue. This is important since individual animals are being investigate and there will be inter-tumor and inter-individual variation. For the RT-PCR data, amplification of an internal control is required since relative band intensities are discussed.

The transcriptomic data are clear and carefully documented but I suggest that the data could be summarized better. e.g. in Figure 1, if the transcriptome profile for the viral genomes from each rabbit was very similar, why not just show Fig1D and add a table to show the % reads for each of the four replicates. There are several repetitions of the viral genome map (Figs 2, 3, 4, 5, 7,8) and the transcriptome of rabbit #4940 (Figs 1,2 & 3). This detail is not required in every figure. For figures where primers or probes are used, tables giving the locations and sequences of the various primers used would be helpful. Naming the primers/probes in the various panels of all the figures e.g. E6 probe1 or E1^E4 primer pairs X etc. would then allow the reader to refer to the primer/probe table.

Figure 7 is unhelpful due to the poor quality of the northern blots. The authors cannot conclude from the data what transcript each band represents, whether bands represent one or multiple transcripts or whether there are the same or different bands between panels. The northern blot data do not add to the transcriptomic and RT-PCR data and should be removed.

The Alphafold predictions in Figure 9 can only be useful if modelled on top of the corresponding viral proteins from other PVs as has been done for CRPVLE6 and HPV16 E6. On top of this, considerably more annotation would be required, together with 3D images. I don’t believe these data, as they stand, add to the manuscript and suggest they be removed.

The western blots in Figure 10 cannot be considered quantitative unless loading controls and replicates are performed and relative quantification of protein expression is carried out. For example, although the overall protein levels are much less in RK13 cells compared to HEK293 cells, it looks like there is at least as much increase in LE6 by inhibiting the proteasome in RK13 cells as in HEK293 cells. In addition, in Figure 10C, I don’t understand the relevance of comparative expression of FLAG-tagged CRPV proteins in the two cell lines especially as there is no internal loading control.

In Figure 11, rabbit and kidney cell lines are used to try to determine the location of expression of the CRPV proteins in cells. This experiment is flawed since it does not take account of the normal host cell of the virus, which is keratinocytes. More importantly, PV gene expression responds to epithelial differentiation. It should be possible to carry out raft culture of primary or immortalized rabbit keratinocytes to address the cell type issue and to properly assess the subcellular location of viral proteins in a tissue, especially the late proteins since they are expressed in differentiated keratinocytes.

Reviewer #3: None

**Part III – Minor Issues: Editorial and Data Presentation Modifications**

Reviewer #1: l. 103-106: More recent studies investigating interactors of E6 should be also described and included (PMID: 20663910; PMID: 18067942, PMID: 36155393)

l.119: it would be more appropriate to mention/reference the PAVE web site for HPV16 and 31 (and other PV) instead of ref. 67 and 68. Ref. 70 has nothing to do with PV transcript maps and needs to be removed.

l. 211/212: the late promoter in E7 was first identified in HPV31 and 16 by the Laimins lab (PMID: 1326657) and the Iftner lab (PMID: 8642661). These references should be included.

l. 227: heterogenous start sites for the HPV31 late promoter were first described in PMID: 9525589. Please add.

l. 347-349: CRPV E5 is present on L2 transcripts and thus might be a late protein. I therefore recommend to include it on the map and modify the sentence accordingly.

CRPV E1, E2, L/SE6, E7, E8^E2, and E10 (E8) have been functionally investigated by several groups. Simple overexpression of these proteins and analysis by immunoblot and immunofluorescence does not really add new information compared to previous work. Therefore, Fig. 9 – 11 and l.433-484 can be deleted from the results section.

I miss a statement and references in the results and/or discussion section that four CRPV promoters and five splice sites have been discovered before by other groups.

Reviewer #2: Why does LE6 staining not go way in the presence of MG132 in Figure 11, as suggested by Fig 10? There is also a lot of nuclear staining for E1^E4 in this figure despite what the authors claim.

The discussion could focus on comparison with other PVs and mention cancer progression.

On lines 423 and 432 “corneal layers” is incorrect, they mean “cornified” layers.

Reviewer #3: 1. Previous work has indicated that the Cottontail Rabbit Papillomavirus is not able to efficiently complete its productive life cycle in domestic laboratory rabbits, but in this study, the authors show evidence of productive genome amplification in the New Zealand White rabbit background. Could the authors add some additional dialogue to help the reader to understand this.

2. It is not always clear which transcripts are ‘major’ transcripts, and which cases are minor species. This is important when considering protein function, particularly for transcripts that map to E2, which is a transcription factor that occurs as forms that drive transcriptional activation and repression of the viral early promotors. If the authors could add some additional commentary, this would be helpful.

PLOS authors have the option to publish the peer review history of their article (what does this mean?). If published, this will include your full peer review and any attached files.

Reviewer #1: No

Reviewer #2: No

Reviewer #3: No
---

## [Decision Letter · Decision Letter 1]

30 Sep 2024

Dear Dr. Zheng,

Thank you very much for submitting your manuscript "THE FULL TRANSCRIPTION MAP OF COTTONTAIL RABBIT PAPILLOMAVIRUS IN TUMOR TISSUES" for consideration at PLOS Pathogens. As with all papers reviewed by the journal, your manuscript was reviewed by members of the editorial board and by several independent reviewers. The reviewers appreciated the attention to an important topic. Based on the reviews, we are likely to accept this manuscript for publication, providing that you modify the manuscript according to the review recommendations.

Sincerely,

Paul F. Lambert

Academic Editor

PLOS Pathogens

Patrick Hearing

Section Editor

PLOS Pathogens

Michael Malim

Editor-in-Chief

PLOS Pathogens

orcid.org/0000-0002-7699-2064

Reviewer Comments (if any, and for reference):

Reviewer's Responses to Questions

**Part I - Summary**

Reviewer #2: This study provides a comprehensive description of the transcriptome and coding potential of the Hershey cotton tail rabbit papillomavirus. It adds to the literature on papillomavirus gene expression.

Reviewer #3: The previous reviewer’s comments were generally supportive of the study, and appreciated the fact that this study significantly advances the usefulness of the CRPV model. The authors have clarified the text, and where requested, have improved the quality of the figures to make them easier to understand. It is by far he most comprehensive analysis of CRPV gene expression that we have. The work is presented to a high standard, and is a valuable addition to the field that I am happy to recommend for publication.

**Part II – Major Issues: Key Experiments Required for Acceptance**

Reviewer #2: (No Response)

Reviewer #3: NA

**Part III – Minor Issues: Editorial and Data Presentation Modifications**

Reviewer #2: The authors have revised their manuscript according to the reviewers’ comments. The work is now more robust and more easily understood in relation to work on other papillomaviruses. The new alphafold figure is particularly helpful.

I still have concerns regarding the northern blots. The images are improved but, in general, the resolution of northern blots does not allow multiple transcripts of similar sizes to be definitively assigned to bands. For example, the description of Blot 1 does not correspond to the text “As shown in Fig 6B, Fig 5C, and S5 Table, Probe 120 immediately downstream of P90 detected three distinct products, the most abundant ~1.9-kb transcript D with single nt 1371^3714 splicing and less ~2.2-kb transcript G with nt 1751^3714 splicing, ~2.5-kb transcript E with nt 1371^3065 splicing,….”

The text states that three products are seen but goes on to describe four mRNAs. The most abundant transcript looks to be greater than 2kb, not 1.9 kb as stated. Is it possible to be certain that there are two bands around 2 kb considering the resolution in this part of the gel? The band representing transcript E at ~2.5 kb cannot be seen in this blot.

It seems that the authors are assuming that the bands in the blots, e.g. the major band at ~2 kb, represent multiple transcripts but due to the resolution of the gels, it should be made clear in the text that the bands were assigned by presumption from the transcriptomic and RT-PCR data.

Reviewer #3: NA

PLOS authors have the option to publish the peer review history of their article (what does this mean?). If published, this will include your full peer review and any attached files.

Reviewer #2: No

Reviewer #3: No

Figure Files:

Data Requirements:

Reproducibility:

References:

---

## [Editor Report · Decision Letter 2]

8 Oct 2024

Dear Dr. Zheng,

We are pleased to inform you that your manuscript 'THE FULL TRANSCRIPTION MAP OF COTTONTAIL RABBIT PAPILLOMAVIRUS IN TUMOR TISSUES' has been provisionally accepted for publication in PLOS Pathogens.

Best regards,

Paul F. Lambert

Academic Editor

PLOS Pathogens

Patrick Hearing

Section Editor

PLOS Pathogens

Michael Malim

Editor-in-Chief

PLOS Pathogens

orcid.org/0000-0002-7699-2064
---

## [Editor Report · Acceptance letter]

18 Oct 2024

Dear Dr. Zheng,

We are delighted to inform you that your manuscript, "THE FULL TRANSCRIPTION MAP OF COTTONTAIL RABBIT PAPILLOMAVIRUS IN TUMOR TISSUES," has been formally accepted for publication in PLOS Pathogens.

Best regards,

Michael Malim

Editor-in-Chief

PLOS Pathogens

orcid.org/0000-0002-7699-2064